# Unified Transferability Metrics for Time Series Foundation Models

**Weiyang Zhang[1]   Xinyang Chen[1✉]   Xiucheng Li[1]   Kehai Chen[1]   Weili Guan[2]   Liqiang Nie[1]**

[1]School of Computer Science and Technology, Harbin Institute of Technology (Shenzhen)
[2]School of Information Science and Technology, Harbin Institute of Technology (Shenzhen)
{zhangweiyang12138, chenxinyang95, nieliqiang}@gmail.com
{lixiucheng, chenkehai, guanweili}@hit.edu.cn

## Abstract

With the increasing number of time series pre-trained models, designing transferability evaluation metrics for time series has become an urgent problem to address. While transferability evaluation has been extensively studied in computer vision, we aim to address a critical gap by developing tailored metrics for time series analysis. In this paper, we introduce TEMPLATE, a transferability estimation framework specifically tailored for versatile time series analysis, comprising three complementary metrics: (1) Dependency Learning Score quantifies a model's capacity to capture temporal dependencies. (2) Pattern Learning Score evaluates the representation quality in extracting discriminative temporal patterns. (3) Task Adaptation Score assesses cross-task generalization capability, enabling versatile time series analysis. TEMPLATE presents a versatile framework compatible with both classification and regression paradigms. Through comprehensive benchmarking across 5 distinct downstream tasks, our method demonstrates superior capability in identifying optimal pre-trained models from heterogeneous model pools for transfer learning. Compared to the state-of-the-art method ETran, our approach improves the weighted Kendall's $\tau_w$ across 5 downstream tasks by 35%. The code is available at https://github.com/TEMPLATE.

## 1   Introduction

Recently, pre-trained models have drawn increasing attention in the time series domain due to their exceptional performance in computer vision and natural language processing [1]. These models have achieved significant success across various time series downstream tasks [2, 3] and are readily available on platforms like HuggingFace [4] and TensorFlow Hub [5]. However, no single model consistently outperforms others across all datasets. Therefore, selecting the most suitable pre-trained time series model for a given target task has become a pressing challenge. A time-consuming solution is to fine-tune all pre-trained models on the target dataset and then select the best-performing fine-tuned model. But compared to pre-trained models in computer vision such as ResNet [6] and MobileNet [7], time-series pre-trained models have much larger parameter scales [8, 9], making direct brute-force fine-tuning incur enormous time costs and high computational resource requirements [10], as shown in the left part of Figure 1.

Recent studies propose fast transferability evaluation methods to efficiently rank models and select the optimal one. Existing methods can generally be categorized into static and dynamic approaches [11]. Static methods calculate scores directly based on the statistical information of the model, such as LEEP [12], NLEEP [13], H-score [14] and TMI [15]. In contrast, dynamic methods transform this statistical information using certain learning frameworks or representation space mapping algorithms before calculating scores, such as SFDA [16], LogME [17], and ETran [18]. These approaches are empirically validated as effective metrics for selecting computer vision models.

39th Conference on Neural Information Processing Systems (NeurIPS 2025).

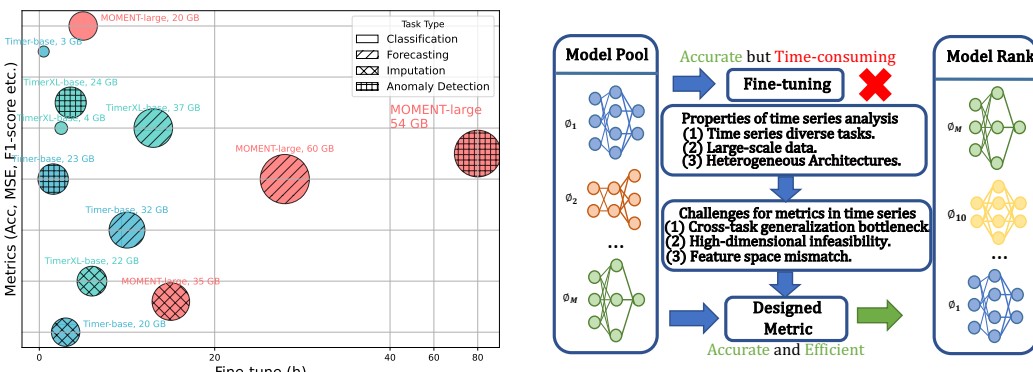

Figure 1: Pre-trained models' fine-tune overhead (left) and transferability metric framework (right).

Nevertheless, existing research primarily focuses on image-based tasks. When applied to time series, several challenges emerge, as shown in the right part of Figure 1. First, mainstream time series tasks exhibit significant diversity, encompassing forecasting, classification, imputation, and anomaly detection, among others. While existing methods predominantly concentrate on classification-based tasks, achieving cross-task generalization capability remains the primary challenge. Second, downstream time series datasets frequently exhibit large-scale characteristics, resulting in high-dimensional computational infeasibility. This consequently imposes stringent requirements on evaluation metrics, demanding both computational efficiency and measurement accuracy. Last but not least, computer vision models consistently employ standardized data preprocessing protocols, consequently maintaining minimal variance in feature dimensionality. But in time series analysis, the architectural heterogeneity of models and their divergent approaches to processing inter-channel dependencies lead to substantial discrepancies in feature dimensionality. This consequently induces feature space mismatch, rendering evaluation metrics that rely on numerical magnitudes such as intra-class variance [15, 19] or singular value [17] ineffective. Table 1 summarizes the challenges in developing transferability metrics for time series. Prior to this paper, there is no ideal solution for pre-trained model selection specially designed to tackle these challenges.

To address these fundamental challenges while accounting for the intrinsic characteristics of time series, we propose TEMPLATE (TEMPoraL representAtion Transferability Estimation), which quantitatively assesses the transferability of time series pre-trained models through three complementary evaluation metrics: Firstly, an effective time series pre-trained model must demonstrate robust capability in capturing and representing temporal dependencies. Moreover, we empirically demonstrate that the principal components constitute the most transferable elements within learned representations. Consequently, we propose the Dependency Learning Score as a quantitative metric to assess dependency learning capability through comparative analysis of principal components between original representations and their corresponding representations in trend. Secondly, to assess the core ability of time series pre-trained models in capturing key temporal patterns, we introduce the Pattern Learning Score as a quantitative evaluation metric. This score precisely measures the model's focus on key temporal patterns through the ratio of the principal component to the overall pattern complexity. Finally, to ensure adaptability across diverse downstream tasks and enable versatile time series analysis, we introduce the Task Adaptation Score as a task-specific evaluation metric for assessing representation quality. For high-level tasks, hierarchical representations are preferred. Thus we calculate the similarity between the features output by the first and last layers of the encoder. Through the integration of these three complementary evaluation metrics, we establish a comprehensive framework for assessing time series pre-trained models. Experimental results demonstrate strong correlation between our proposed metric and actual model transferability, consistently outperforming existing evaluation methods in benchmark tests across 5 downstream tasks.

Our contributions are summarized as follows:

- We evaluated the challenges in designing transferability metrics for time series and, from the model's dependency learning ability, pattern learning ability, and task adaptation ability, proposed the TEMPLATE pre-trained model transferability evaluation framework.

Table 1: Time series design challenges and methods comparison.

| Challenges | NCE | LEEP | LogME | SFDA | NLEEP | PACTran | TMI | ETran | NCTI | Ours |
|---|---|---|---|---|---|---|---|---|---|---|
| Cross-task generalization bottleneck | ✗ | ✗ | ✓ | ✗ | ✗ | ✗ | ✗ | ✓ | ✗ | ✓ |
| High-dimensional infeasibility | ✗ | ✗ | ✗ | ✗ | ✗ | ✗ | ✗ | ✓ | ✓ | ✓ |
| Feature space mismatch | ✗ | ✗ | ✗ | ✗ | ✗ | ✗ | ✗ | ✗ | ✗ | ✓ |

- Our proposed framework is highly flexible and generalizable, supporting both classification and regression tasks. Moreover, the metrics can be applied to 5 mainstream time series analysis tasks, demonstrating our broad applicability and task-specific adaptability.
- Experiments on 32 datasets across 5 mainstream time-series tasks demonstrate that our method achieves state-of-the-art (SOTA) performance in evaluating the transferability of time-series pre-trained models, with an average improvement of 35% compared to previous SOTA methods.

## 2 Related Work

### 2.1 Time Series Pre-trained Model

In the image and vision domains, pre-training on large-scale sequence datasets has significantly advanced modality understanding [20, 21]. Building on this, pre-trained models have been widely developed in the time series domain [1]. LLM4TS [22] enhances forecasting accuracy by fine-tuning large language models, while Tempo [23] improves cross-modal alignment using multimodal prompts. Lag-Llama [24] conducts pre-training on multi-domain time series data and uses lag features as covariates for probabilistic univariate forecasting. Moment [25] conducts masked reconstruction pre-training on various time series datasets. Similarly, Timer [9] undergoes autoregressive pre-training on large-scale collected datasets. UniTS [26] integrates downstream tasks into a single framework through task tokenization. Time-MOE [27], which combines the mixture-of-experts method, is the first to scale time series pre-trained models to 2.4 billion parameters. Therefore, as the scale of time-series pre-trained models continues to grow, how to evaluate the transferability of these models has become an urgent issue that needs to be addressed.

### 2.2 Model Transferability Estimation

Model Transferability Estimation (MTE) aims to provide fast and efficient methods for quantifying a model's performance on downstream tasks without fine-tuning [11]. Recent MTE advancements focus on vision models. Early works, such as NCE [28] and LEEP [12], focused on probabilistic methods based on the expected empirical distribution of target labels. LogME [17] estimates the maximum marginalized likelihood of labels, while NLEEP [13] extends LEEP by replacing the output layer with a Gaussian Mixture Model. SFDA [16] projects features into Fisher space and enhances class separability using physics-inspired models. TMI [15] uses intra-class feature variance as a performance indicator, assuming that lower variance reflects tighter clustering of class features. NCTI [19], inspired by the neural collapse phenomenon, develops metrics to measure the distance between the current state of the pre-trained model and its hypothetical fine-tuned state. ETran [18] introduces energy scores to quantify whether the target dataset is in-distribution or out-of-distribution for a candidate model, assuming models with higher in-distribution levels for the target dataset exhibit greater transferability. Existing studies primarily focus on vision and have not fully considered the characteristics of time-series data and pre-trained models. Therefore, we propose TEMPLATE, a transferability evaluation framework specifically designed for time-series pre-trained models.

## 3 Approach

### 3.1 Problem Formulation

Given $M$ pretrained models $\{\phi_m\}_{m=1}^M$ and a target dataset $\mathcal{D}$ with $N$ samples, where each model consists of a backbone $f$ that outputs encoded features, and the target dataset is associated with an evaluation metric (accuracy, MSE, F-score etc.), we initialize a predictor head on the backbone $f$ and fine-tune the entire model on the target dataset. The feature extracted by the $l$-th layer of the pretrained model $\phi_m(\cdot)$ is denoted as $\mathbf{H}^l$, where $\mathbf{H}^l = \phi_m(\mathbf{X}) \in \mathbb{R}^{N \times d}$ and $d$ is the feature dimension. In the

following, we omit $l$ using $\mathbf{H}$ to denote the feature output of the last layer of the model's encoder. By performing brute-force fine-tuning on all models, we obtain the true performance $\{P_m\}_{m=1}^M$ for the model hub. A practical model selection method should generate a score for each pretrained model. Ideally, the score should correlate strongly with the actual fine-tuning performance $\{P_m\}_{m=1}^M$, allowing the best-performing pretrained model to be selected based on the evaluation score alone.

## 3.2 Transferability Assessment through Temporal Representation Transferability Estimation

To effectively evaluate the transferability of time series pre-trained models, it is necessary to address the dimensional gap in feature matrices caused by the model's different handling of inter-channel relationships, as well as the varying requirements of various downstream tasks. Therefore, we avoid directly using statistical quantities of the feature matrix to measure the transferability of pre-trained models. Specifically, we propose TEMPLATE, a hybrid transferability metric that includes Dependency Learning Score, Pattern Learning Score, and Task Adaptation Score, aiming to assess transferability of the learned representation in pre-trained models from the perspective of temporal dependency learning, temporal pattern learning and hierarchical features.

To better understand how fine-tuning affects the representation learned by pre-trained models, we explore the changes in the components of feature matrices before and after fine-tuning. Specifically, we use Singular Value Decomposition (SVD) to analyze this process as it efficiently decomposes the feature matrix and analyzes its dominant patterns. We fine-tune MOMENT-small [25], Timer-small [9], and UniTS-small [26] on EthanolConcentration [29], Handwriting [29], and UWaveGestureLibrary [29] dataset, respectively, and compare the cosine similarity of the feature vectors corresponding to singular values before and after fine-tuning. The results are shown in Figure 2 and it is observed that the cosine similarity of feature vectors decreases as their corresponding singular values decrease. This indicates that larger singular values encapsulate more knowledge and are more transferable to downstream tasks.

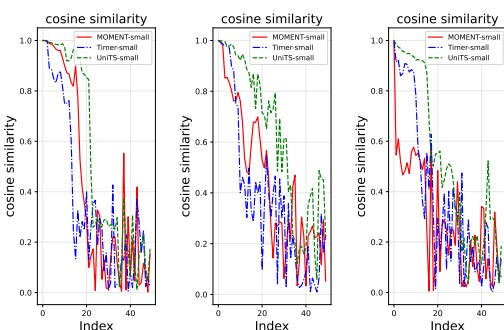

Figure 2: The cosine similarity of feature vectors, computed based on feature vectors before and after fine-tuning, on three datasets (sorted in descending order by singular values): EthanolConcentration (Left), Handwriting (Middle), and UWaveGestureLibrary (Right).

**Dependency Learning Score.** The core of time series modeling is to capture temporal dependency. Time series pre-trained models should be able to capture long-term dependencies, which represent stable and persistent patterns crucial for effective modeling [30]. To assess this ability, we employ series decomposition [31] to extract the trend component, which inherently contains long-term dependencies. This component is fed into the model, yielding a feature matrix $\mathbf{T}$ that is focused on capturing long-term dependencies, as follows:

$$\text{trend}(\mathbf{X}) = \text{AvgPool}(\text{padding}(\mathbf{X})), \mathbf{T} = \phi_m(\text{trend}(\mathbf{X})), \tag{1}$$

where we use padding to maintain the original series length, and then apply the AvgPool layer for moving average calculations. A model proficient in learning long-term dependencies should exhibit a high degree of consistency between the feature matrix $\mathbf{H}$ and the feature matrix $\mathbf{T}$ of the trend component. To this end, we propose the Dependency Learning Score $S_{dl}$. Specifically, considering that the largest singular value and its corresponding feature vector demonstrate superior transferability, we prioritize the use of the feature vectors corresponding to the largest singular values of $\mathbf{H}$ and $\mathbf{T}$. We perform SVD to decompose the features:

$$\mathbf{H} = \mathbf{U}_h \Sigma_h \mathbf{V}_h^T, \mathbf{T} = \mathbf{U}_t \Sigma_t \mathbf{V}_t^T. \tag{2}$$

We denote the largest singular values of $\mathbf{H}$ and $\mathbf{T}$ as $\sigma_h$ and $\sigma_t$, and their corresponding feature vectors as $u_h$ and $u_t$, respectively. With the feature vectors, $S_{dl}$ is formulated to quantify the pre-trained model's capability in capturing long-term temporal dependencies through computation of the Pearson correlation coefficient [32] between the $u_h$ and $u_t$, as follows:

$$S_{dl} = \rho(u_h, u_t) = \frac{\text{Cov}(u_h, u_t)}{\lambda_h \lambda_t}, \tag{3}$$

where $\mathrm{Cov}(\cdot)$ represents the covariance between feature vectors, and $\lambda_h$, $\lambda_t$ represents their respective variances. Since $S_{dl}$ represents the pre-trained model's ability to learn long-term dependencies, a pre-trained model that effectively learns long-term dependencies should achieve a larger $S_{dl}$.

**Pattern Learning Score.** Effective temporal pattern learning is critical for time series pre-trained models to handle complex, multi-scale behaviors, which allows the model to learn the temporal dynamics in time series data, thereby achieving superior performance on downstream tasks.

To this end, we propose the Pattern Learning Score $S_{pl}$ to measure the pre-trained model's ability to learn the primary temporal patterns. Within the framework of matrix decomposition, singular values quantitatively characterize the relative significance of distinct patterns embedded in the feature matrix. The largest singular value typically corresponds to the primary temporal pattern in time series, reflecting the most significant dynamic behavior captured by the model.

While the dominant singular value provides an intuitive measure of a model's capacity to capture primary temporal patterns, the inherent feature space mismatch resulting from heterogeneous inter-channel relationship processing across pre-trained models may significantly compromise the reliability of transferability assessment. Therefore, we introduce the nuclear norm as a measure of the overall importance of the patterns, defined as the sum of all singular values of the matrix. Specifically, we use the ratio of the largest singular value to the nuclear norm to measure the pre-trained model's ability to learn primary temporal patterns. A higher ratio signifies enhanced capability of the pre-trained model in capturing dominant temporal patterns, whereas a lower ratio potentially indicates insufficient representation of these critical temporal patterns. Moreover, compared to the raw data, the temporal patterns of the trend component are simpler, with the primary patterns being more prominent. Therefore, the ratio of the largest singular value to the nuclear norm using the trend component provides a more accurate evaluation of the pre-trained model's ability to learn primary temporal patterns. The formulation of $S_{pl}$ is as follows:

$$S_{pl} = \frac{\sigma_t}{\|\mathbf{T}\|_*}, \tag{4}$$

where $\sigma_t$ denotes the largest singular value of the trend component feature matrix, and $\|\mathbf{T}\|_*$ represents its nuclear norm. Since $S_{pl}$ represents the pre-trained model's ability to learn primary temporal patterns, a pre-trained model that effectively learns these patterns should achieve a larger $S_{pl}$.

**Task Adaptation Score.** Time series downstream tasks exhibit substantial heterogeneity, with distinct tasks imposing markedly divergent requirements on feature representations [33]. Classification tasks require high-level semantics, and imputation tasks require the model to learn from missing data. Consequently, hierarchical representations become essential. In contrast, forecasting tasks and anomaly detection tasks place more emphasis on the precise extraction of low-level features from the raw data itself, focusing on local patterns and temporal dependencies [34, 33]. Consequently, an effectively designed pre-trained model should demonstrate the capability to learn task-adaptive representations that dynamically adjust to varying downstream task requirements. [35]. Moreover, due to the diverse designs of time-series pre-trained models and the significant differences in network architectures, it is necessary to consider the overall evolution of features. We measure the task adaptation ability of the pretrained model by calculating the similarity between the output features $\mathbf{H}^1$ from the first layer of the encoder and the output features $\mathbf{H}$ from the last layer. A simple approach is to directly compute the dot product-based similarity [36] between them:

$$\langle \mathrm{vec}(H^1(H^1)^T), \mathrm{vec}(HH^T) \rangle = \mathrm{tr}(H^1(H^1)^T HH^T) = \|H^T H^1\|_F^2, \tag{5}$$

Due to the differences in how time series pre-trained models handle inter-channel relationships, there is a significant gap in feature matrix dimensions across different pre-trained models. Directly calculating squared dot product can lead to inaccurate estimates of model's transferability. Therefore, we use centered kernel alignment [37, 38, 36] to measure similarity, which further optimizes dot product-based similarity. Specifically, Equation 5 means that, for the centered $\mathbf{H}^1$ and $\mathbf{H}$:

$$\frac{1}{(n-1)^2} \mathrm{tr}(H^1(H^1)^T HH^T) = \|\mathrm{cov}((H^1)^T, H^T)\|_F^2, \tag{6}$$

The Hilbert-Schmidt Independence Criterion [39] generalizes Equations 5 and 6 to inner products from reproducing kernel Hilbert spaces, where the squared Frobenius norm of the cross-covariance matrix

becomes the squared Hilbert-Schmidt norm of the cross-covariance operator. Let $K_{ij} = k(\mathbf{x}_i, \mathbf{x}_j)$ and $L_{ij} = l(\mathbf{y}_i, \mathbf{y}_j)$ where $k$ and $l$ are two kernels. The empirical estimator of HSIC is:

$$\text{HSIC}(K, L) = \frac{1}{(n-1)^2} \text{tr}(K\mathbf{Q}L\mathbf{Q}), \tag{7}$$

where $\mathbf{Q}$ is the centering matrix $\mathbf{Q}_n = I_n - \frac{1}{n}\mathbf{1}\mathbf{1}^{\text{T}}$. Finally, we normalize the HSIC to obtain our score function, $S_{ra}$:

$$S_{ta} = \frac{\text{HSIC}(K, L)}{\sqrt{\text{HSIC}(K, K)\text{HSIC}(L, L)}}, \tag{8}$$

where we use the RBF kernel $k(\mathbf{x}_i, \mathbf{x}_j) = \exp(-||\mathbf{x}_i - \mathbf{x}_j||_2^2/(2\gamma^2))$ for mapping, with several possible strategies for selecting the bandwidth. This strategy controls the extent to which similarity at small distances is emphasized over large distances. We set $\gamma$ to the median distance between samples. For classification and imputation task, it is essential for the model to learn hierarchical representations, making a lower $S_{ta}$ more desirable. In contrast, for forecasting and anomaly detection task, the model needs to focus on capturing fundamental features, making a higher $S_{ta}$ more appropriate.

**Overall Score Function.** In this paper, we expect pre-trained models to learn valuable temporal dependencies and patterns from the time series data and output task adaptive ability tailored to different downstream tasks, enabling versatile time series analysis. To this end, we design three score functions: $S_{dl}$, $S_{pl}$, and $S_{ta}$. $S_{dl}$ and $S_{pl}$ measure the dependency learning ability of the model and primary temporal pattern learning, respectively. $S_{ta}$ evaluates the task-adaptive ability of the pre-trained model to various downstream tasks. Since time series often involves large-scale datasets, directly applying SVD can result in high computational costs. To accelerate the computation, we use the power iteration method [40, 41] to approximate the largest singular value of the model and its corresponding eigenvector. For detailed information about the power iteration method, please refer to Appendix C.4. Secondly, we quickly calculate the nuclear norm using equation 9, as shown below:

$$\|\mathbf{T}\|_* = \text{tr}\left(\sqrt{(\mathbf{T})^T\mathbf{T}}\right). \tag{9}$$

By employing these two methods, we avoid the high computational cost associated with SVD, enabling a fast evaluation of the transferability of pre-trained models. In addition, since the score functions have different scales, directly summing these scores may negatively impact the performance of transferability evaluation. Therefore, we normalize each score to a unit range (0 to 1) instead of manually defining hyperparameters to balance the contributions of each score, as follows:

$$S_{dl} \leftarrow \frac{S_{dl} - \min(S_{dl})}{\max(S_{dl}) - \min(S_{dl})}. \tag{10}$$

Similarly, we can obtain the normalized scores $S_{pl}(\mathbf{T})$ and $S_{ta}(\mathbf{H}^1, \mathbf{H})$. By combining these three scores, we derive the final transferability evaluation metric:

$$S_{total} = S_{dl}(\mathbf{H}, \mathbf{T}) + S_{pl}(\mathbf{T}) + \lambda S_{ta}(\mathbf{H}^1, \mathbf{H}), \tag{11}$$

where $\lambda$ is determined by the downstream task type. When evaluating on classification task and imputation task, $\lambda$ is set to -1. For forecasting task and anomaly detection task, $\lambda$ is set to 1. Note that the three scores are equally weighted, and $\lambda$ acts as a sign function determined by the task type, without requiring fine-tuning for specific downstream dataset. The pre-trained model with a higher overall score $S_{total}$ indicates better transferability within the model pool for the target dataset $\mathcal{D}$.

## 4 Experiments

In this section, we examine the effectiveness of TEMPLATE in assessing the transfer learning performance of pretrained models across 5 mainstream downstream time series tasks, including classification, short- and long-term forecasting, imputation, and anomaly detection.

**Implementation Details.** To quantify the correlation between estimated assessment scores and actual fine-tuning results, we employ the weighted Kendall's $\tau_w$ (see Appendix C.3), which measures ranking agreement with higher weights for higher ranks. More details about the dataset and experiment implementation can be found in Appendix C.

Table 2: Comparison of SOTA Rate and Relative Improvement for Different Methods.

| Method | GBC [42] | CC-FV [43] | NCTI [19] | ETran [18] | Ours |
|---|---|---|---|---|---|
| SOTA Rate in Original Paper | 37% | 36% | 40% | 45% | **46%** |
| Improvement (vs Previous SOTA) | 20% | 9% | 21% | 18% | **35%** |

**Pre-trained Models.** We select 10 pre-trained models from 4 model families to form the pre-trained model pool, including MOMENT-small [25], MOMENT-base [25], MOMENT-large [25], Timer-small [9], Timer-base [9], TimerXL-small [44], TimerXL-base [44], UniTS-small [26], UniTS-base [26], and UniTS-large [26]. We fine-tune all source models on the target dataset to obtain the actual ranking. More details about the pre-trained model pool can be found in the AppendixC.2.

## 4.1 Main Results.

As a transferability evaluation metric designed for pre-trained models in time series, TEMPLATE achieves consistently state-of-the-art accuracy across 5 mainstream analytical tasks, as shown in Figure 3. Notably, no method has achieved SOTA on all dataset, and as illustrated in Table 2, our method attains a remarkably high SOTA rate. We show more experimental results in Appendix D to further demonstrate the effectiveness of the method.

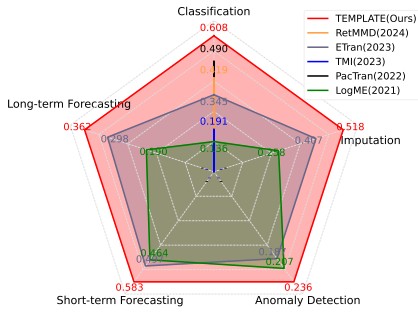

Figure 3: Performance comparison.

## 4.2 Classification

**Setup.** Time series classification can be applied to tasks such as recognition and medical diagnosis [45]. Specifically, We verify all methods on 9 multivariate datasets from the UEA classification archive [29], covering gesture, audio recognition, etc., across multiple task categories. We preprocess the datasets according to [46], where different subsets have varying sequence lengths.

Table 3: Classification Benchmark: Performance (Weighted Kendall's $\tau_w$) of different methods. The best results are highlighted in **bold** while the second best are underlined.

| Datasets | LogME (2021) | NLEEP (2022) | SFDA (2022) | PACTran (2022) | TMI (2023) | NCTI (2023) | ETran (2023) | RetMMD (2024) | TEMPLATE (Ours) |
|---|---|---|---|---|---|---|---|---|---|
| EthanolConcentration | 0.567 | 0.432 | 0.120 | 0.488 | -0.430 | -0.32 | 0.686 | 0.512 | **0.724** |
| FaceDetection | -0.203 | 0.092 | **0.598** | 0.306 | -0.600 | 0.109 | -0.359 | 0.310 | 0.597 |
| Handwriting | -0.445 | -0.104 | 0.314 | 0.596 | 0.768 | 0.700 | 0.478 | 0.365 | **0.822** |
| JapaneseVowels | 0.231 | 0.213 | 0.021 | 0.306 | 0.302 | 0.340 | -0.196 | **0.654** | 0.447 |
| PEMS-SF | -0.472 | -0.612 | -0.312 | 0.306 | -0.300 | 0.053 | 0.076 | **0.520** | 0.470 |
| SelfRegulationSCP1 | 0.356 | 0.529 | 0.459 | 0.619 | 0.300 | 0.310 | **0.651** | 0.450 | 0.484 |
| SelfRegulationSCP2 | 0.268 | 0.241 | -0.198 | 0.457 | 0.455 | 0.450 | **0.667** | 0.397 | 0.551 |
| SpokenArabicDigits | 0.342 | 0.321 | -0.367 | **0.744** | 0.647 | -0.210 | 0.479 | 0.201 | 0.637 |
| UWaveGestureLibrary | 0.584 | 0.127 | 0.440 | 0.592 | 0.576 | 0.245 | 0.624 | 0.362 | **0.719** |
| Average | 0.136 | 0.138 | 0.119 | 0.490 | 0.191 | 0.186 | 0.345 | 0.419 | **0.608** |

**Performance Comparison.** We compare our method with existing transferability evaluation metrics, including LogME [17], NLEEP [13], SFDA [16], PACTran [47], TMI [15], NCTI [19], ETran [18] and RetMMD [48]. Table 3 shows the results of our method compared to previous work on classification benchmarks. Our method outperforms all previous methods, achieving a state-of-the-art (SOTA) $\tau_w$ of 0.608, which is 24% ahead of the second-best method. Due to the different ways models handle inter-channel relationships, feature-statistics-based methods, such as LogME [17], TMI [15], and NCTI [19], generally perform poorly on datasets with a large number of channels, like FaceDetection and PEMS-SF. In contrast, our method achieves outstanding results on these two datasets, with $\tau_w$ of 0.597 and 0.470, respectively. These results highlight the effectiveness of our approach in evaluating pre-trained time series models for transfer learning.

## 4.3 Short- and Long-term Forecasting

**Setup.** Time series forecasting plays a crucial role in areas such as weather forecasting, traffic management, and energy consumption. To thoroughly evaluate the generalizability of our approach in forecasting, we employ two types of benchmarks: long-term forecasting and short-term forecasting. For long-term forecasting, we use seven widely recognized long-term time series forecasting datasets [31], covering practical applications across various domains. For short-term forecasting, we adopt the M4 dataset [49], which includes univariate marketing data collected annually, quarterly, and monthly.

Table 4: Long-term Forecasting Benchmark: All the results are averaged from 4 different prediction lengths. '-' means out of memory. See Table 12 in Appendix for the full results.

| Methods | ETTh1 | ETTh2 | ETTm1 | ETTm2 | Weather | Electricity | Traffic | Average |
|---------|-------|-------|-------|-------|---------|-------------|---------|---------|
| LogME | 0.215 | 0.167 | 0.400 | 0.565 | 0.114 | -0.130 | - | 0.190 |
| ETran | 0.138 | **0.212** | 0.189 | 0.351 | **0.474** | 0.302 | 0.192 | 0.265 |
| Ours | **0.240** | -0.003 | **0.518** | **0.576** | 0.412 | **0.361** | **0.432** | **0.362** |

**Performance Comparison.** Tables 4 and 11 present the experimental results of our method in long-term and short-term forecasting scenarios. We compare our method with existing transferability evaluation metrics for regression tasks, including LogME [17] and ETran [18]. The core of long-term forecasting task lies in the model's ability to effectively capture complex temporal dependencies and long-term trend. By contrast, short-term forecasting task focus more on the precise capture of fine-grained temporal patterns. Overall, our method demonstrates outstanding performance across both scenarios. Notably, LogME performs poorly on multi-channel datasets like Electricity and cannot evaluate the performance of pre-trained models on Traffic due to out of memory issues. In contrast, our method achieves excellent results on both datasets. These results further validate the effectiveness of our method in assessing time series pre-trained models for transfer learning.

## 4.4 Imputation

**Setup.** Real-time systems are continuously operating and monitored by automated observation devices. However, due to failures, the collected time series may be partially missing, makes downstream analysis challenging. Therefore, imputation has become widely used in practical applications. In this paper, we select datasets from electricity, weather, and traffic scenarios as benchmarks.

Table 5: Imputation Benchmark: All results are averaged from 4 mask ratios. '-' means out of memory. See Table 13 in Appendix for the full results.

| Methods | ETTh1 | ETTh2 | ETTm1 | ETTm2 | Weather | Electricity | Traffic | Average |
|---------|-------|-------|-------|-------|---------|-------------|---------|---------|
| LogME | 0.322 | 0.238 | 0.607 | 0.388 | 0.312 | -0.061 | - | 0.258 |
| ETran | 0.324 | **0.419** | 0.254 | 0.310 | 0.441 | **0.456** | 0.398 | 0.372 |
| Ours | **0.406** | 0.317 | **0.664** | **0.650** | **0.645** | 0.412 | **0.529** | **0.518** |

**Performance Comparison.** Effective time series imputation places high demands on the model's ability to capture temporal dependencies and understand contextual information. Specifically, when imputing missing values, the model needs to utilize the implicit temporal patterns within data, including modeling dynamic relationships between time points and accurately inferring both the overall trends and local features of time series. Table 5 presents the experimental results of our method in the time series imputation scenario. In the experiments, we evaluate the model's performance under different missing ratios. The results show that, whether in scenarios with a low or high proportion of missing data, our method demonstrates a high level of model transferability evaluation accuracy.

## 4.5 Anomaly Detection

**Setup.** Anomaly detection in sensor data is crucial for industrial maintenance, as anomalies are often hidden within large-scale datasets, making data labeling challenging. We compare 5 widely used anomaly detection benchmarks: SMD [50], MSL [51], SMAP [51], SWaT [52], and PSM [53], covering applications in service monitoring, spatial data, earth exploration, and water treatment.

Table 6: Anomaly Detection Benchmark: Performance (Weighted Kendall's $\tau_w$) of different methods. The best results are highlighted in **bold** and '-' means out of memory.

| Methods | PSM | MSL | SMAP | SMD | SWaT | Average |
|---|---|---|---|---|---|---|
| LogME | 0.230 | 0.468 | 0.340 | - | - | 0.207 |
| ETran | **0.433** | -0.024 | 0.121 | **0.126** | **0.277** | 0.187 |
| Ours | 0.142 | **0.515** | **0.431** | -0.08 | 0.173 | **0.236** |

**Performance Comparison.** Effective time series anomaly detection requires the model to identify normal and abnormal patterns in the data and accurately distinguish between the two. Table 6 presents the results of our method in the time series anomaly detection scenario. The experimental results show that our method effectively evaluates the model's transferability in anomaly detection scenarios.

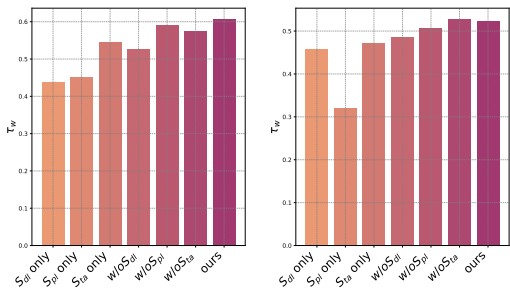

Figure 4: Effectiveness of component in TEM-PLATE on Classification and Imputation task.

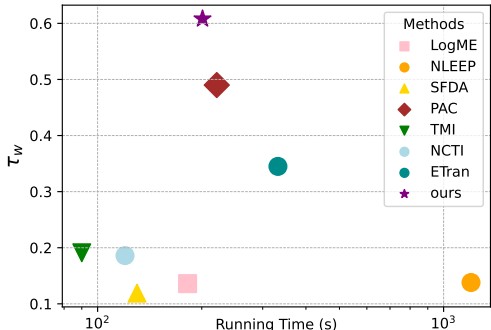

Figure 5: Time complexity analysis.

## 4.6 Analysis

**Ablation Study.** Our TEMPLATE consists of three sub-items designed to evaluate the performance of pre-trained models on downstream tasks. To explore the individual contribution of each sub-item to the final metric and analyze its impact on overall performance, we conduct an ablation study. Figure 4 presents the results of the ablation study on the classification and imputation task, with the complete results available in Appendix D.2. It can be observed that all three metrics achieve positive ranking correlations, and their combination yields the highest average ranking correlation, indicating that the three metrics complement each other and can comprehensively measure the performance of pre-trained models.

**Time Complexity.** In the Figure 5, we present a comparison of the average runtime and $\tau_w$ on the classification task across 9 datasets using different model selection metrics. While NCTI [19] and TMI [15] are fast, they perform poorly in evaluating the transferability of time series pre-trained models due to their reliance on statistical metrics of feature matrices. LogME [17] is efficient, but it cannot be applied to large datasets like Traffic and SMD because it relies on the magnitude of singular values. In contrast, our method uses the power iteration method to avoid high computational costs and enabling its application to large datasets. Although ETran [18] efficiently obtains the $S_{en}$ score, the LDA-based score calculation incurs significant time costs, resulting in slower overall speed.

## 5 Conclusion

In this paper, we propose TEMPLATE, a pre-trained model evaluation framework specifically designed for time series. We are the first to design a transferability evaluation metric tailored for time series. TEMPLATE consists of three scores, aimed at measuring the transferability of pre-trained models from two perspectives: dependency and pattern learning, and task adaptation. Our approach has been successfully applied to 5 mainstream time series downstream tasks. Extensive experiments demonstrate that our method achieves state-of-the-art (SOTA) performance in evaluating time series pre-trained models for transfer learning.

**Acknowledgements**

This work was supported by the National Natural Science Foundation of China (62306085, 62206074, 62476071, U23B2055, U24A20328), Shenzhen College Stability Support Plan (GXWD20231130151329002, GXWD20220811173233001), CCF-ALIMAMA TECH Kangaroo Fund (CCF-ALIMAMA OF 2025001), Guangdong Basic and Applied Basic Research Foundation (2025A1515012932, 2025A1515011732), Shenzhen Science and Technology Program (KQTD20240729102154066, ZDSYS20230626091203008).

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

# A  Limitations

Our method may perform poorly on some datasets with severe distribution shifts or structural breaks, as the results on the training set cannot accurately reflect the model's performance on the test set. This remains a challenging problem, and no existing method has effectively addressed it yet.

# B  Broader impacts

The work in this paper contributes to facilitating the evaluation of transfer capability for time series pre-trained models, thereby further saving time and training resources and enabling the rapid selection of suitable time series pre-trained models for downstream tasks.

# C  Experiment Details

## C.1  Implement Details

We provide detailed descriptions of the datasets in Tables 8. For all 5 downstream tasks, we follow the experimental setup of [34]. To compute the transfer performance values, we carefully fine-tuned the pre-trained models through hyperparameter grid search. As [54] highlighted, learning rate and weight decay are the two most critical parameters. Therefore, we performed grid search over learning rates and weight decay values (6 learning rates ranging from $10^{-3}$ to $10^{-5}$, and 3 weight decay values from $10^{-3}$ to $10^{-5}$) to select the optimal hyperparameters. The fine-tuning experiments of the pre-trained models were conducted on an NVIDIA H20 GPU with 96GB of memory. It is important to note that we only fine-tuned the pre-trained models to validate the effectiveness of the proposed method; the method itself does not require fine-tuning of the pre-trained models. All the results of pre-trained model transferability evaluation metrics were obtained on an AMD EPYC 7513 32-Core CPU.

## C.2  Model Pool Details

We selected 10 pre-trained models from four model families to form the model pool, including MOMENT-small [25], MOMENT-base [25], MOMENT-large [25], Timer-small [9], Timer-base [9], TimerXL-small [44], TimerXL-base [44], UniTS-small [26], UniTS-base [26], and UniTS-large [26]. As shown in Table 7, the selected pre-trained models differ in structure, model parameters, and the scale of pre-training. The MOMENT models were reconstructed and trained on the large hybrid dataset Time-series Pile; Timer-small and TimerXL-small were pre-trained on the ERA5 dataset, while Timer-base and TimerXL-base were pre-trained on the large dataset UTSD; the UniTS series models were pre-trained on a multi-domain hybrid dataset for time series. This highlights the differences between the models and further demonstrates that the proposed method is sufficient to support general transferability.

Table 7: The Architecture, Model size, and Scale of the pre-trained data of the model pool.

| Method | Timer | Timer-XL | MOMENT | UniTS |
|---|---|---|---|---|
| Architecture | Decoder | Decoder | Encoder | Encoder |
| Model size | 11M~67M | 14M~69M | 40M~380M | 3M~8.4M |
| Pre-training Scale | 0.5B, 28B | 0.5B, 28B | 1.13B | 0.5B |

## C.3  Evaluation Metric

To evaluate the effectiveness of the proposed method, we refer to previous work [17, 18] and use Kendall's $\tau$ [55] to measure the correlation between the estimated evaluation scores and the ground-truth ranking scores. This metric assesses ranking consistency, where a larger $\tau$ value indicates greater effectiveness in ranking models. It is defined as the number of concordant pairs minus the

number of discordant pairs, divided by the overall number of pairs $\binom{M}{2}$, as follows:

$$\tau = \frac{2}{M(M-1)} \sum_{1 \leq i < j \leq M} \text{sgn}(G_i - G_j).\text{sgn}(T_i - T_j). \tag{12}$$

Furthermore, we adopt a weighted version of Kendall's $\tau$ [56], denoted as $\tau_w$, to assign higher weights to models ranked higher, to quantify this correlation.

### C.4 Power Iteration

Due to the limited space in the main text, we show the detailed procedure of the power iteration method here, as shown in Algorithm 1:

---
**Algorithm 1** Power Iteration
---

**Input:** Matrix $\mathbf{H}$, initial vector $v_0$, number of iterations $k$, here we set 10.
**Output:** Dominant eigenvalue $\lambda_{\max}$, corresponding eigenvector $v_k$
Initialize $v_0$ randomly
**for** $i = 1$ **to** $k$ **do**
    $v_{i+1} = \mathbf{H}v_i$
    Normalize $v_{i+1}$ to unit length
**end for**
Estimate the dominant eigenvalue $\lambda_{\max} = \frac{v_k^T \mathbf{H} v_k}{v_k^T v_k}$

---

## D  More Experimental Results

### D.1  Statistical Significance Testing

To validate the effectiveness of TEMPLATE over ETran, we conducted a statistical significance test. The results show that the mean of TEMPLATE (0.483) is significantly higher than that of ETran (0.359), with a mean difference of 0.124, which is statistically significant (T_statistic = 2.679, P_value = 0.012). Therefore, TEMPLATE demonstrates significantly better performance across all tasks compared to ETran, confirming its effectiveness improvement.

### D.2  Ablation Study

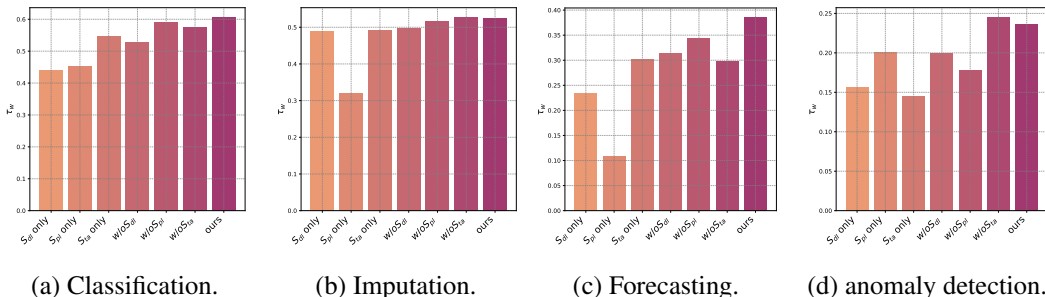

|          (a) Classification. | (b) Imputation. | (c) Forecasting. | (d) anomaly detection. |

Figure 6: Effectiveness of each individual component in TEMPLATE. We use the three terms individually or remove them one at a time from the full system. The results are averaged across all datasets.

Due to space limitations, we have included the complete ablation experiment results in Figure 6. The experimental results show that, compared to the three individual metrics, TEMPLATE achieves the best performance, demonstrating the effectiveness of TEMPLATE. Additionally, we have also validated the effectiveness of combining these metrics in pairs. The combination of $S_{dl}$ and $S_{pl}$ outperforms each metric individually, proving their complementary nature. This demonstrates that evaluating the pre-trained model's ability for time series semantic learning from the perspectives of

Table 8: Dataset descriptions. The dataset size is organized in (Train, Validation, Test).

| Tasks | Dataset | Dim | Series Length | Dataset Size | Information (Frequency) |
|---|---|---|---|---|---|
| Forecasting (Long-term) | ETTm1, ETTm2 | 7 | {96, 192, 336, 720} | (34465, 11521, 11521) | Electricity (15 mins) |
| | ETTh1, ETTh2 | 7 | {96, 192, 336, 720} | (8545, 2881, 2881) | Electricity (15 mins) |
| | Electricity | 321 | {96, 192, 336, 720} | (18317, 2633, 5261) | Electricity (Hourly) |
| | Traffic | 862 | {96, 192, 336, 720} | (12185, 1757, 3509) | Transportation (Hourly) |
| | Weather | 21 | {96, 192, 336, 720} | (36792, 5271, 10540) | Weather (10 mins) |
| Forecasting (short-term) | M4-Yearly | 1 | 6 | (23000, 0, 23000) | Demographic |
| | M4-Quarterly | 1 | 8 | (24000, 0, 24000) | Finance |
| | M4-Monthly | 1 | 18 | (48000, 0, 48000) | Industry |
| | M4-Weakly | 1 | 13 | (359, 0, 359) | Macro |
| | M4-Daily | 1 | 14 | (4227, 0, 4227) | Micro |
| | M4-Hourly | 1 | 48 | (414, 0, 414) | Other |
| Imputation | ETTm1, ETTm2 | 7 | 96 | (34465, 11521, 11521) | Electricity (15 mins) |
| | ETTh1, ETTh2 | 7 | 96 | (8545, 2881, 2881) | Electricity (15 mins) |
| | Electricity | 321 | 96 | (18317, 2633, 5261) | Electricity (15 mins) |
| | Traffic | 862 | 96 | (12185, 1757, 3509) | Transportation (Hourly) |
| | Weather | 21 | 96 | (36792, 5271, 10540) | Weather (10 mins) |
| Classification (UEA) | EthanolConcentration | 3 | 1751 | (261, 0, 263) | Alcohol Industry |
| | FaceDetection | 144 | 62 | (5890, 0, 3524) | Face (250Hz) |
| | Handwriting | 3 | 152 | (150, 0, 850) | Handwriting |
| | JapaneseVowels | 12 | 29 | (270, 0, 370) | Voice |
| | PEMS-SF | 963 | 144 | (267, 0, 173) | Transportation (Daily) |
| | SelfRegulationSCP1 | 6 | 896 | (268, 0, 293) | Health (256Hz) |
| | SelfRegulationSCP2 | 7 | 1152 | (200, 0, 180) | Health (256Hz) |
| | SpokenArabicDigits | 13 | 93 | (6599, 0, 2199) | Voice (11025Hz) |
| | UWaveGestureLibrary | 3 | 315 | (120, 0, 320) | Gesture |
| Anomaly Detection | SMD | 38 | 100 | (566724, 141681, 708420) | Server Machine |
| | MSL | 55 | 100 | (44653, 11664, 73729) | Spacecraft |
| | SMAP | 25 | 100 | (108146, 27037, 427617) | Spacecraft |
| | SWaT | 51 | 100 | (396000, 99000, 449919) | Infrastructure |
| | PSM | 25 | 100 | (105984, 26497, 87841) | Server Machine |

dependency learning and pattern learning is an effective and reasonable approach. Although $S_{ta}$ did not achieve the best correlation in the anomaly detection task, compared to W/O $S_{ta}$, TEMPLATE showed an 8% average improvement in $t_w$ across 5 downstream tasks, effectively proving the validity of $S_{ta}$.

Additionally, it is important to note that TEMPLATE is a very robust method. We conducted additional ablation experiments to explore the specific implementation methods for the scores. For $S_{dl}$, the main idea is to measure the alignment of the eigenvector corresponding to the largest singular value between the trend component and the original sequence. In the table, we also implemented $S_{dl}$ based on Euclidean distance, and it can be seen that $S_{dl}$ based on Euclidean distance achieves a positive transferability evaluation. For $S_{ta}$, the main idea is to measure the similarity between features

Table 9: Ablation study on the specific implementation methods of $S_{dl}$ and $S_{ta}$.

| Method | EthanolConcentration | Handwriting | PEMS - SF | FaceDetection |
|---|---|---|---|---|
| $S_{dl}$(euclidean_distance) | 0.332 | 0.597 | 0.370 | 0.361 |
| $S_{dl}$(pearsonr) | 0.349 | 0.608 | 0.381 | 0.366 |
| $S_{ta}$(dot product ) | 0.421 | 0.654 | 0.276 | 0.450 |
| $S_{ta}$(HSIC) | 0.436 | 0.698 | 0.350 | 0.453 |

to meet the requirements of different task types for features. In the following table, we implemented $S_t$ based on squared dot product, and it can be observed that $S_{ta}$ based on squared dot product also achieves positive transferability evaluation. HSIC maps the features to a higher-dimensional space and then computes the similarity. The method of measuring feature similarity has good properties [36]. Therefore, we use HSIC to measure similarity. The experimental results in the table further demonstrate the effectiveness and robustness of TEMPLATE.

Table 10: Ablation Experiment on the Sensitivity of Hyperparameter $\lambda$.

| $\lambda$ | EthanolConcentration | Handwriting | PEMS - SF | SelfRegulationSCP2 |
|---|---|---|---|---|
| -0.6 | 0.716 | 0.822 | 0.461 | 0.551 |
| -0.8 | 0.724 | 0.822 | 0.461 | 0.551 |
| -1 | 0.724 | 0.822 | 0.470 | 0.551 |

Finally, we note that the three scores in the final score are equally important, and $\lambda$ is only related to the task type rather than the specific downstream task dataset. To demonstrate the robustness of our method, we add a parameter sensitivity experiment for $\lambda$ in Table 10. The experimental results show that on specific downstream datasets, there is no need to fine-tune $\lambda$.

Table 11: Short-Term Benchmark: Performance (Weighted Kendall's $\tau_w$) of different methods. The best results are highlighted in **bold**.

| Datasets | Yearly | Quarterly | Monthly | Others | Average |
|---|---|---|---|---|---|
| LogME | 0.492 | 0.332 | 0.562 | **0.468** | 0.464 |
| ETran | 0.235 | **0.763** | 0.624 | 0.318 | 0.485 |
| Ours | **0.534** | 0.667 | **0.712** | 0.417 | **0.583** |

### D.3 Full Result

Due to space limitations in the main text, we have included the results of the short-term forecasting, long-term forecasting and imputation in Tables 11, Tables 12 and 13. The experimental results demonstrate that compared to LogME and ETran, TEMPLATE achieves the best average performance.

Table 12: The full result for forecasting task.

| Datasets | ETTh1 | | | | ETTh2 | | | | ETTm1 | | | | ETTm2 | | | | weather | | | | ECL | | | | Traffic | | | |
|---|---|---|---|---|---|---|---|---|---|---|---|---|---|---|---|---|---|---|---|---|---|---|---|---|---|---|---|---|
| Length | 96 | 192 | 336 | 720 | 96 | 192 | 336 | 720 | 96 | 192 | 336 | 720 | 96 | 192 | 336 | 720 | 96 | 192 | 336 | 720 | 96 | 192 | 336 | 720 | 96 | 192 | 336 | 720 |
| LogME | 0.169 | 0.375 | 0.198 | 0.121 | 0.103 | 0.234 | 0.132 | 0.199 | 0.596 | 0.232 | 0.364 | 0.411 | 0.522 | 0.516 | 0.700 | 0.522 | 0.310 | 0.021 | -0.110 | 0.233 | 0.031 | -0.332 | 0.060 | -0.280 | - | - | - | - |
| ETran | -0.007 | 0.276 | 0.282 | 0.001 | 0.321 | 0.003 | 0.180 | 0.345 | 0.473 | 0.126 | 0.314 | -0.154 | 0.210 | 0.393 | 0.312 | 0.489 | 0.531 | 0.642 | 0.384 | 0.341 | 0.461 | 0.184 | 0.296 | 0.268 | 0.102 | 0.168 | 0.314 | 0.184 |
| Ours | 0.234 | 0.309 | 0.243 | 0.175 | -0.012 | 0.076 | 0.131 | -0.199 | 0.746 | 0.317 | 0.670 | 0.340 | 0.438 | 0.598 | 0.680 | 0.589 | 0.496 | 0.523 | 0.342 | 0.291 | 0.180 | 0.346 | 0.458 | 0.458 | 0.642 | 0.334 | 0.395 | 0.361 |

Table 13: The full result for imputation task.

| Datasets | ETTh1 | | | | ETTh2 | | | | ETTm1 | | | | ETTm2 | | | | weather | | | | ECL | | | | Traffic | | | |
|---|---|---|---|---|---|---|---|---|---|---|---|---|---|---|---|---|---|---|---|---|---|---|---|---|---|---|---|---|
| Mask Ratio | 12.5% | 25% | 37.5% | 50% | 12.5% | 25% | 37.5% | 50% | 12.5% | 25% | 37.5% | 50% | 12.5% | 25% | 37.5% | 50% | 12.5% | 25% | 37.5% | 50% | 12.5% | 25% | 37.5% | 50% | 12.5% | 25% | 37.5% | 50% |
| LogME | 0.193 | 0.441 | 0.251 | 0.405 | 0.214 | 0.242 | 0.196 | 0.301 | 0.342 | 0.613 | 0.658 | 0.815 | 0.417 | 0.365 | 0.423 | 0.348 | 0.402 | 0.195 | 0.297 | 0.351 | 0.0390 | -0.150 | -0.091 | -0.051 | - | - | - | - |
| ETran | 0.287 | 0.384 | 0.340 | 0.283 | 0.383 | 0.391 | 0.451 | 0.451 | 0.190 | 0.266 | 0.169 | 0.392 | 0.255 | 0.305 | 0.245 | 0.436 | 0.414 | 0.432 | 0.556 | 0.360 | 0.580 | 0.345 | 0.298 | 0.601 | 0.284 | 0.497 | 0.309 | 0.501 |
| Ours | 0.323 | 0.340 | 0.477 | 0.485 | 0.283 | 0.291 | 0.408 | 0.287 | 0.606 | 0.536 | 0.774 | 0.741 | 0.645 | 0.479 | 0.543 | 0.935 | 0.596 | 0.645 | 0.678 | 0.662 | 0.368 | 0.319 | 0.611 | 0.351 | 0.428 | 0.623 | 0.471 | 0.593 |

