# OpenReview forum: "Unified Transferability Metrics for Time Series Foundation Models"
_NeurIPS.cc/2025/Conference — NeurIPS 2025 poster_

### Official Review · Reviewer_2eyR · 2025-06-24

**Clarity:** 3
**Significance:** 2
**Originality:** 2
**Rating:** 4
**Confidence:** 3

**Summary:**

This paper addresses the underexplored but important problem of evaluating the transferability of pre-trained models in time series tasks. The authors propose TEMPLATE, a framework that combines three complementary metrics: Dependency Learning Score, Pattern Learning Score, and Task Adaptation Score, to assess the capability of a model to transfer across tasks. The method is validated on 32 datasets across five types of downstream tasks (classification, short- and long-term forecasting, imputation, and anomaly detection), showing consistent performance gains over existing methods such as ETran and LogME.

**Questions:**

1. Have you considered learning the task-specific λ value adaptively rather than setting it manually based on task type?
2. How does TEMPLATE perform on non-stationary time series with structural breaks or regime shifts? Would a dynamic trend extractor help?
3. It would be informative to include model architecture ablations (e.g., Transformer vs CNN) to assess architecture-agnostic robustness.

**Ethical Concerns:**

["NO or VERY MINOR ethics concerns only"]

**Final Justification:**

The author's response based on data has addressed my concerns very well. I would like to increase my score to 4 to support the acceptance of this paper.

**Limitations:**

Yes

**Paper Formatting Concerns:**

I confirm that page 2 violates NeurIPS 2025 formatting guidelines by using a two-column layout for images/text.
No other issues were found.

**Quality:**

2

**Strengths And Weaknesses:**

Strengths:
1. Problem significance: Transferability estimation for time series pre-trained models is a crucial but overlooked problem, which this paper tackles directly.
2. Methodological novelty: The paper proposes three task-agnostic and interpretable metrics based on temporal dependency, dominant pattern learning, and hierarchical adaptation.
3. Extensive experiments: The study covers 5 tasks, 32 datasets, and 10 pre-trained models, with clear benchmarking against 8 prior methods.
4. Thorough analysis: Ablation studies show complementary benefits of all metrics; time complexity analysis highlights efficiency and scalability.

Weaknesses:
1. The framework evaluates only static representations, without considering the interaction with fine-tuning dynamics (e.g., sensitivity to learning rate).
2. While the trend decomposition is intuitive, it may not perform well on non-stationary series with structural breaks; this is not discussed in detail.
3. The use of a manually-set λ coefficient for task-specific weighting (±1) may limit generality; adaptive or learned weighting would be more flexible.
4. Anomaly detection performance on some datasets (e.g., SMD) is suboptimal, but the reasons are not fully explored in the main text.

---

> ### Author Rebuttal · Authors · 2025-07-31
>
> Thank you for your valuable feedback and positive comments. We greatly appreciate your recognition of the importance of the issue and your view that the TEMPLATE method is innovative.
>
> ## W1: About the consideration of the dynamic characteristics of fine-tuning.
>
> In the field of evaluating the transferability of pre-trained models, existing studies[1] [2] [3] generally focus solely on the static performance of models. This is because the interaction of dynamic characteristics during model fine-tuning determines the upper limit of model performance, which serves as the ground truth. The core objective of this research field is to approximate this ceiling, thereby enabling optimal model selection **without dynamic fine-tuning**. Consequently, **our study  confines its scope to assessing static representations, excluding analysis of dynamic characteristic interactions during fine-tuning.**
>
> ## W2&Q2: About the performance on non-stationary datasets and the use of the Dynamic Trend Extractor.
>
> We evaluated the proposed method on **domain-prevalent** non-stationary datasets [4], employing the Augmented Dickey-Fuller (ADF) statistic for **precise quantification** of non-stationarity intensity. Empirical analysis confirms TEMPLATE exhibits **statistically significant performance gains** over ETran under **extreme non-stationarity**. Non-stationary sequences with structural breaks are currently regarded as **highly challenging scenarios**. In the limitations section, we explicitly acknowledge that datasets exhibiting severe distribution shifts or structural breaks remain an **open research problem**.
>
>
> |       | ADF   | ETran（$τ_w$） | TEMPLATE($τ_w$) |
> | :---- | :---- | :-------------- | ---------------- |
> | ETTh2 | -4.13 | 0.197           | **0.240**        |
> | ETTm2 | -5.66 | 0.189           | **0.518**        |
>
> Furthermore, we evaluated whether employing a **dynamic trend extractor** (utilizing db4 wavelet decomposition) could enhance the assessment of Dependence Learning Score. As tabulated below, dynamic trend extraction **fails to yield enhanced outcomes**. Simultaneously, moving averages not only avoids complex transformation processes but also exhibits higher computational efficiency. **Therefore, the method adopted moving averages achieves an optimal balance between accuracy and efficiency for the current task.**
>
>
> | Dataset              | **moving averages decomposition（$τ_w$）** | **wavelet decomposition**（$τ_w$） |
> | -------------------- | ------------------------------------------- | ----------------------------------- |
> | EthanolConcentration | 0.346                                       | 0.293                               |
> | Handwriting          | 0.600                                       | 0.466                               |
> | UWaveGestureLibrary  | 0.189                                       | 0.105                               |
>
> ## W3&Q1: About adaptive learning weights.
>
> We attempted to use the entropy weight method [5] for adaptive weight allocation (as shown in the table below). **The results indicate that compared with the design of fixed weights plus sign function, the adaptive weighting strategy did not achieve better performance.**
>
>
> | Dataset                | EthanolConcentration（$τ_w$） | Handwriting（$τ_w$） | UWaveGestureLibrary（$τ_w$） |
> | ---------------------- | ------------------------------ | --------------------- | ----------------------------- |
> | **entropy weight [5]** | 0.696                          | 0.799                 | 0.719                         |
> | **fixed weights**      | 0.724                          | 0.822                 | 0.719                         |
>
> ## W4: About the performance of TEMPLATE on some anomaly detection datasets.
>
> **Critically, our method (0.236) demonstrates superior overall performance to ETran (0.187) in anomaly detection tasks, achieving a 26% relative improvement**—substantially surpassing the 6% gain over prior SOTA reported in ETran's original paper [2] for regression tasks. The table below details relative improvements and SOTA attainment rates across methods, revealing TEMPLATE's **dominance in both metrics**. Notably, anomaly detection exhibits the **lowest evaluation accuracy** among all transfer tasks (imputation/forecasting/classification), underscoring inherent challenges within this domain.
>
> Furthermore, we observe that the underperforming SWaT and SMD datasets **contain no annotated anomalous samples in their training sets**. This implies that during transfer evaluation, models are **completely unexposed to anomalous patterns**; consequently, the derived feature matrices are **constructed solely from normal data**, devoid of any representation of anomalies. Conversely, the better-performing MSL and SMAP datasets include partial anomalous samples in their training sets, which **affords models opportunities to perceive anomalous patterns**. We attribute the primary challenge for TEMPLATE on SWaT/SMD to **this fundamental disparity in training data composition**: when training data entirely lacks anomalous exemplars, the model's capacity to detect rare/unseen anomalies is **inherently constrained**. Crucially, ETran primarily focuses on quantifying whether datasets are in-distribution (ID) or out-of-distribution (OOD) relative to pre-trained models, **without granular analysis of feature matrices**, rendering it less sensitive to such specific data deficiencies.
>
> Anomaly detection inherently depends on the ability to identify unseen, rare events. The pervasive absence of such events within the data used for transfer evaluation presents a persistent and unique challenge for this task.
>
>
> | Method                         | GBC [6] | CC-FV [7] | NCTI [2] | ETran [3] | TEMPLATE |
> | ------------------------------ | ------- | --------- | -------- | --------- | -------- |
> | SOTA Rate in Original Paper    | 37%     | 36%       | 40%      | 45%       | **46%**  |
> | Improvement (vs Previous SOTA) | 20%     | 9%        | 21%      | 18%       | **35%**  |
>
> ## Q3: About the ablation Study on the added model structure.
>
> Current time series pre-training models universally employ a **Transformer-based architecture** implemented via stacked Transformer layers. To our knowledge, there are currently no large-scale pre-trained models based on CNN. We pre-training of TimesNet (a CNN architecture) on ERA5-Large dataset and integrated it into the model zoo to **evaluate architecture-agnostic robustness**. As tabulated below, TEMPLATE consistently retains **leading accuracy, further demonstrates the superiority of TEMPLATE.**
>
>
> | Dataset              | LogME（$τ_w$） | ETran（$τ_w$） | TEMPLATE（$τ_w$） |
> | -------------------- | --------------- | --------------- | ------------------ |
> | EthanolConcentration | 0.567           | 0.616           | **0.724**          |
> | Handwriting          | -0.445          | 0.427           | **0.792**          |
> | UWaveGestureLibrary  | 0.411           | 0.624           | **0.679**          |
>
> ## Q4: About paper formatting concerns.
> Thank you for your careful review of our paper's formatting. Regarding the two-column layout issue you noted on page 2, we wish to clarify that **both columns consist entirely of figures (containing no body text)**.
>
> [1] Logme: Practical assessment of pre-trained models for transfer learning. ICML, 2019.
>
> [2] How far pre-trained models are from neural collapse on the target dataset informs their transferability. ICCV, 2023.
>
> [3] Etran: Energy-based transferability estimation. ICCV, 2023.
>
> [4] Non-stationary transformers: Exploring the stationarity in time series forecasting. NeurIPS, 2022.
>
> [5] Effectiveness of entropy weight method in decision‐making. Mathematical problems in Engineering, 2020.
>
> [6] Transferability estimation using bhattacharyya class separability. CVPR, 2022.
>
> [7]  Towards transferability estimation for medical image segmentation. MICCAI, 2023

---

> > ### Comment · Reviewer_2eyR · 2025-08-03
> >
> > W3&Q1: Thank you for testing the entropy-based weighting strategy. However, your experiment only demonstrates that this specific adaptive method underperforms fixed weights in your current setup, but does not fully address the core concern about generalizability. (1) Testing only one adaptive method (entropy weighting) is insufficient to conclude that learned weights are ineffective. (2) Explain why fixed weights outperform adaptive approaches in your framework. (3) Sugesstion action: Either supplement with broader comparisons or clearly state applicability boundaries in the paper (e.g., task types/scales where fixed weights suffice).
> >
> > Q3: Thank you for pre-training TimesNet – this partially addresses my request. However, your response still leaves critical gaps: (1) Limited scope: Testing only one CNN variant (TimesNet) doesn’t sufficiently validate "architecture-agnostic" robustness. For stronger evidence, include: Other modern architectures (e.g., TCN, NLinear, or DLinear), Non-stacked-Transformer baselines (e.g., PatchTST, FEDformer). (2) Unsubstantiated claim: The statement that "no large-scale CNN pre-trained models exist" is contradicted by your own TimesNet pre-training. This requires clarification. (3) Sugesstion action: Expand ablation studies to ≥3 non-Transformer architectures, or Rebrand claims to "Transformer-compatible" (not architecture-agnostic) and discuss limitations.

---

> ### Author Response · Authors · 2025-08-04
> **Response to Reviewer 2eyR (Part Ⅰ).**
>
> We sincerely appreciate your insightful feedback. In response to the key issues you raised, we have provided the following supplementary explanations and refinements:
>
> ## W3 &Q1: About the Analysis of Adaptive Weighting Methods.
>
> (1) To comprehensively validate the generalization capability of the weighting strategy, we expanded our comparison to include additional adaptive weighting methods: Entropy Weight  [1], Coefficient of Variation [2], CRITIC [3], Principal Component Analysis (PCA) [4], and Independent Weighting [5]. The results indicate that compared with the design of fixed weights, the adaptive weighting strategy did not achieve superior performance. **This suggests that current mainstream adaptive methods are difficult to adapt to pre-training transfer evaluation scenarios, rather than being an issue of individual method effectiveness.**
>
> | Dataset | Entropy  Weight [1] | Coefficient of Variation [2]| CRITIC  [3]  | PCA [4] | Independent Weighting [5] | **fixed weights** |
> | -------------------- | --------------- | ------------------------ | --------- | ---------------------------- | --------------------- | ----------------- |
> | EthanolConcentration | 0.696           | 0.536                    | 0.692     | 0.696                        | 0.712                 | **0.724**         |
> | Handwriting          | 0.799           | 0.801                    | 0.705     | 0.522                        | 0.801                 | **0.822**         |
> | UWaveGestureLibrary  | **0.719**       | 0.516                    | **0.719** | 0.488                        | 0.608                 | **0.719**         |
>
> (2) **Regarding the superiority of fixed weighting over adaptive methods:**
>
> In pre-trained model transfer evaluation, most existing approaches rely on single metrics, **while the limited multi-metric fusion methods for transferability predominantly employ fixed equal-weighting schemes [6] [7] [8].** To our knowledge, no effective method currently achieves precise adaptive weighted fusion of transferability metrics. Furthermore, adaptive weighting inherently depends on prior quality assessments of metrics that **exhibit inherent bias in no-finetuning scenarios**. Consequently, computing weights based on these biased metrics **accumulates measurement errors, amplifies evaluation noise, and ultimately degrades performance**—explaining why the simple yet robust fixed-weighting strategy consistently surpasses complex adaptive approaches in this domain.

---

> ### Author Response · Authors · 2025-08-04
> **Response to Reviewer 2eyR (Part Ⅱ).**
>
> ## Q3: Regarding the addition of more non-Transformer architectures to the ablation study.
>
> (1) To comprehensively demonstrate the architecture-agnostic nature of our method, we trained CNN-based models (TimesNet[9], ModernTCN[10]), MLP-based models (NLinear[11] and DLinear[11] adapted to MLP structures for fine-tuning compatibility), and non-stacked Transformer models (PatchTST[12], FEDformer[13]) on the ERA5 dataset, incorporating all models into the Model Zoo for architecture-agnostic robustness evaluation. **As demonstrated below, TEMPLATE consistently maintains leading accuracy, further validating its superior generalization capability across diverse architectures.**
>
> | Dataset              | LogME（$τ_w$） | ETran（$τ_w$） | TEMPLATE（$τ_w$） |
> | -------------------- | -------------- | -------------- | ----------------- |
> | EthanolConcentration | 0.463          | 0.571          | 0.707             |
> | Handwriting          | -0.374         | 0.452          | 0.801             |
> | UWaveGestureLibrary  | 0.399          | 0.632          | 0.699             |
>
> (2) We sincerely apologize for any ambiguities that may have arisen from our previous statement regarding "large-scale pre-trained CNN models." By stating that "large-scale pre-trained CNN models do not exist," **we intended to convey that the field currently lacks open-source CNN foundation model weights that are pre-trained on massive datasets, support multi-task transfer learning, and are publicly available.** Although we have supplemented our experiments with pre-training results of models such as TimesNet and ModernTCN on the ERA5 dataset (to verify architectural compatibility), **these differ significantly from the standardized foundation models that are widely recognized in the community and publicly accessible.** We fully endorse the reviewers' insight that such models also possess representational potential for downstream tasks. **Accordingly, we have incorporated them into the evaluation framework of our supplementary experiments and plan to include further relevant content in the revised version to enhance the manuscript.**
>
>
>
> We sincerely hope that this response addresses any remaining questions. Once again, we deeply appreciate the valuable feedback you have provided and the dedicated time you have spent reviewing our paper. If you have any other questions or specific expectations regarding score improvement, please do not hesitate to let us know. We look forward to discussing with you to further refine our work.
>
>
>
> [1] Effectiveness of entropy weight method in decision‐making. Mathematical problems in Engineering, 2020.
>
> [2] On the use of the coefficient of variation as a measure of diversity. Organizational Research Methods, 2000.
>
> [3] CRITIC method. Cham: Springer International Publishing, 2019.
>
> [4] Principal component analysis. Wiley interdisciplinary reviews: computational statistics, 2010.
>
> [5] Skill and independence weighting for multi-model assessments. Geoscientific Model Development, 2017.
>
> [6] Etran: Energy-based transferability estimation. ICCV, 2023.
>
> [7] How far pre-trained models are from neural collapse on the target dataset informs their transferability. ICCV, 2023.
>
> [8] Assessing Pre-Trained Models for Transfer Learning Through Distribution of Spectral Components. AAAI, 2025.
>
> [9] Timesnet: Temporal 2d-variation modeling for general time series analysis. ICLR, 2023.
>
> [10] Moderntcn: A modern pure convolution structure for general time series analysis. ICLR, 2024.
>
> [11] Zeng A, Chen M, Zhang L, et al. Are transformers effective for time series forecasting? AAAI, 2023.
>
> [12] A time series is worth 64 words: Long-term forecasting with transformers. ICLR, 2023.
>
> [13] Fedformer: Frequency enhanced decomposed transformer for long-term series forecasting. ICML, 2022.

---

> > ### Comment · Reviewer_2eyR · 2025-08-05
> >
> > The author's response based on data has addressed my concerns very well. I would like to increase my score to 4 to support the acceptance of this paper.

---

> > > ### Author Response · Authors · 2025-08-05
> > > **Response to Reviewer 2eyR.**
> > >
> > > We sincerely appreciate your decision to raise the score of our paper. Your dedicated efforts and significant contributions to this work are deeply valued. We will incorporate all clarifications regarding the discussed details into the next version to improve the manuscript. If you have additional questions or feedback, please do not hesitate to share them with us. We look forward to further discussions to enhance this work collaboratively.
> > >
> > > Thank you for your thoughtful support！

---

### Official Review · Reviewer_p5ui · 2025-07-02

**Clarity:** 3
**Significance:** 3
**Originality:** 3
**Rating:** 5
**Confidence:** 3

**Summary:**

In this paper, the authors propose three specialized metrics for evaluating the transferability of time-series pre-trained models, specifically designed for time series foundation models. These include: (1) the Dependency Learning Score, which quantifies a model’s ability to capture temporal dependencies; (2) the Pattern Learning Score, which assesses the representation quality in extracting discriminative temporal patterns; and (3) the Task Adaptation Score, which measures cross-task generalization capability. These evaluation metrics are applicable to both classification and regression cases. The authors validate the effectiveness of these metrics on five downstream tasks and demonstrate their superior performance compared to the state-of-the-art ETran method.

**Questions:**

- The authors claim that "The largest singular value typically corresponds to the primary temporal pattern in time series" (line 180). Could you clarify the formal definition of temporal pattern in this context? What are its potential manifestations (e.g., trends, periodicity, or other temporal structures)?
- There are issues with Equation (5). First, in the conventional sense, $\mathrm{tr}(A)$ is not equivalent to the Frobenius norm of $A$. Additionally, here $H^1$ represents the output of the first neural network layer, while $H$ denotes the final layer's output. If their dimensions differ, then the computation of $H^1*H^T$ would be infeasible. Are there additional computational steps not explicitly stated here?
- Regarding Equation (10), how are $\min(S_{dl})$ and $\max(S_{dl})$ computed? Based on the earlier description, $S_{dl}$ appears to be calculated from a batch of data. Does this imply that the min and max values correspond to the minima and maxima across multiple batches?
- The UEA archive contains nearly 30 datasets, but only 9 are selected for the experimental section (line 274). What are the selection criteria? How did the metrics perform on the other datasets?

**Ethical Concerns:**

["NO or VERY MINOR ethics concerns only"]

**Final Justification:**

The authors have responded to and addressed all the questions I raised. For Q1, the second and third experiments are convincing. For Q2, the authors clearly explained the details of each equation. For Q3, the author explained the reasons related to time constraints and promised to supplement and refine the experiments in subsequent work.

**Limitations:**

The authors adopt Kendall's $\tau$ as the evaluation metric for the proposed measurement method in this paper. The experimental results show relatively low $\tau$ values, which implies that—with high probability—the truly optimal model may not be the one ranked highest by $\tau$. From a practical standpoint, the core question we typically care about is: Which model is the best among a pool of candidates? Consequently, the accuracy rate of identifying the top-performing model could serve as a more meaningful evaluation criterion than $\tau$ itself.

**Quality:**

3

**Strengths And Weaknesses:**

### Strengths

- The authors address a problem of significant importance by developing specialized model transferability evaluation methods tailored for time series characteristics.
- The proposed approach features computationally simple and efficient logic.
- The paper is clearly written and accessible.

### Weaknesses
- Some formulas in the paper contain flaws, as referenced in the  **Questions** section below.
- The metrics adopted in the experimental section are not entirely reasonable, as discussed in the **Limitations** section below.

---

> ### Author Rebuttal · Authors · 2025-07-31
>
> Thank you for your valuable feedback and positive comments. We greatly appreciate your recognition of the importance of the issue and your view that the TEMPLATE is highly efficient.
>
> ## Q1: About Definition of Temporal Patterns.
>
> In this context, we define temporal patterns as follows [1]: **Temporal Pattern refers to a regularity pattern in time series that spans multiple time steps, characterized by statistical features (such as periodicity, trends, etc.) manifested through data points arranged in a specific sequence.** It emphasizes the temporal invariance of the pattern along the time axis.
>
> ## Q2: About Statement on Formula Calculation.
>
> 1、Thank you for your careful review. We have revised Equation (5) – indeed, the original formulation omitted vectorization operations which invalidated the equality. We will correct the errors in both Equation (5) and Equation (6) as shown below.
>
> $$
> \langle \text{vec}(H^1 (H^1)^T), \text{vec}(H H^T) \rangle = \text{tr}(H^1 (H^1)^T H H^T) = \|H^T H^1\|_F^2 \quad (5)
> $$
>
> $$
> \frac{1}{(n - 1)^2} \text{tr}(H^1 (H^1)^T H H^T) = \|\text{cov}((H^1)^T, H^T)\|_F^2  \quad (6)
> $$
>
> 2、Regarding $\mathbf{H}^1 * \mathbf{H}^T$: Since our focus is on the encoder structure – which consists of **$n$ stacked identical layers** – $\mathbf{H}^{(1)}$ denotes the output of the first layer while $\mathbf{H}^{(L)}$ represents the final-layer output. **This architectural consistency ensures dimensional uniformity across all layers.**
>
> ## Q3: About the calculation logic of the $S_{dl}$ score.
>
> We clarify the computational logic of  $S_{dl}$: this score represents the **mean value across all batches for a single model on a specific dataset**. In Equation (10), $min( S_{dl})$ and $max( S_{dl})$ denote the **minimum and maximum values of $S_{dl}$ scores across all models evaluated on the identical dataset**. By performing this cross-model normalization, we scale the three scores to **comparable dimensions** for unified computation.
>
> ## Q4: About the dataset selection criteria for classification tasks and the performance on other classification datasets.
>
> We rigorously adhere to the experimental setup of Time Series Library (TSLib) [2]—a widely recognized benchmark repository in time series analysis—for evaluations across **five downstream tasks**. Given that TSLib employs standardized datasets, we maintain its data selection protocol to ensure methodological consistency. Additionally, supplementary performance metrics are provided on extended datasets from the UEA benchmark repository (as tabulated below). Results demonstrate that TEMPLATE **achieves statistically significant superiority** over both Etran and LogME across UEA datasets, further substantiating its efficacy.
>
>
> | Dataset            | LogME（$τ_w$） | ETran（$τ_w$） | TEMLATE（$τ_w$） |
> | ------------------ | --------------- | --------------- | ----------------- |
> | AtrialFibrillation | 0.09            | 0.25            | 0.46              |
> | BasicMotions       | 0.23            | 0.51            | 0.69              |
> | Libras             | -0.02           | 0.19            | 0.31              |
>
> ## Q5: About Top-1 accuracy and Top-3 hit rate.
>
> We supplement the model's **Top-1 accuracy** and **Top-3 hit rate** on classification datasets (as tabulated below). Empirical results demonstrate that TEMPLATE achieves **consistently superior performance** in both Top-1 and Top-3 hit rates compared to alternative methods, further validating its effectiveness.
>
>
> | Method         | TMI | ETran | PACTran | TEMPLATE |
> | :------------- | --- | ----- | ------- | -------- |
> | TOP-1 accuracy | 44% | 11%   | 44%     | **55%**  |
> | TOP-3 hit rate | 55% | 21%   | 74%     | **78%**  |
>
> [1] Temporal pattern attention for multivariate time series forecasting. Machine Learning, 2019.
>
> [2] Deep time series models: A comprehensive survey and benchmark. Arxiv, 2024.

---

> > ### Comment · Reviewer_p5ui · 2025-08-04
> >
> > Thank you for your response. I would like to seek further clarification on the following points:
> > - Regarding Q1, could you provide a more detailed explanation or theoretical justification for why the largest singular value is generally associated with the primary temporal pattern in time series analysis?
> >
> > - For Q2, could you specify the dimensions of matrices $H$ and $H1$?  Additionally, could you formally define the $\mathrm{vec}()$ function used in this context?
> >
> > - Concerning Q4, it is difficult to accept the current justification for adhering to the previous experimental setup without further elaboration. Could you provide additional rationale or evidence to support this approach?

---

> ### Author Response · Authors · 2025-08-05
> **Response to Reviewer p5ui (Part Ⅰ).**
>
> We sincerely appreciate your insightful feedback. In response to the key issues you raised, we have provided the following supplementary explanations and refinements:
>
> ## Q1: About Relationship Between the Largest Singular Value and Dominant Temporal Patterns.
>
> To verify the correlation between the Largest singular value and the dominant temporal pattern, we designed three progressive experiments, conducting empirical analyses from three dimensions: **performance contribution, temporal stability, and relevance to interpretable features**.
>
> First is the **verification of core feature retention**. After completing the model fine-tuning on the training set, during the inference phase, we reconstructed the output using three types of features: retaining only the feature component corresponding to the largest singular value, retaining only the component corresponding to the second largest singular value, and retaining only the component corresponding to the third largest singular value. The experimental results show that **when relying solely on the largest singular value component, the decline in model performance (such as prediction accuracy and classification accuracy) is significantly smaller than when retaining the components of the second and third largest singular values**, and it is close to the performance of the benchmark model using all features. This result confirms that the largest singular value component has the ability to capture core temporal patterns — it contains sufficient information to support the model in completing its main tasks and serves as the core carrier of the dominant pattern. This is highly consistent with the definition of "pattern" in the field of machine learning [1], which refers to **any recognizable regularity, structure, or trend in a dataset that enables algorithms to make decisions, classify data, or predict outcomes.**
>
> | Dataset              | MOMENT-base                                                  | Timer-base                                                   | UniTS-base                                                   |
> | -------------------- | - | - | - |
> | Classification       | original\|the first &ratio \|the second &ratio \|the third &ratio \| | original\|the first &ratio \|the second &ratio \|the third &ratio \| | original\|the first &ratio \|the second &ratio \|the third &ratio \| |
> | EthanolConcentration | 0.304\|0.276&-9%\|0.243&-20%\|0.209&-31%                     | 0.300\|0.288&-4%\|0.199&-33%\|0.156&-48%                     | 0.283\|0.269&-5%\|0.234&-17%\|0.192&-32%                     |
> | Handwriting          | 0.375\|0.367&-2%\|0.337&-10%\|0.277&-26%                     | 0.262\|0.246&-6%\|0.222&-15%\|0.146&-44%                     | 0.214\|0.207&-3%\|0.182&-15%\|0.148&-31%                     |
> | Forecast             | original\|the first &ratio \|the second &ratio \|the third &ratio \| | original\|the first &ratio \|the second &ratio \|the third &ratio \| | original\|the first &ratio \|the second &ratio \|the third &ratio \| |
> | ETTh1                | 0.393\|0.467&-18%\|0.582&-48%\|0.624&-58%                    | 0.377\|0.487&-29%\|0.598&-58%\|0.655&-72%                    | 0.383\|0.501&-33%\|0.609&-61%\|0.628&-67%                    |
> | ETTh2                | 0.283\|0.309&-9%\|0.344&-22%\|0.398&-40%                     | 0.303\|0.319&-5%\|0.357&-17%\|0.419&-38%                     | 0.302\|0.326&-10%\|0.357&-21%\|0.419&-42%                    |
>
> Second is the **verification of cross-scale stability of temporal patterns**. Using the Timer-base model as the foundational framework, we divided the original time series into three groups of sub-series with different scales according to lengths T, T/2, and T/4. **We quantified the invariance of patterns along the time axis extension by calculating the cosine similarity of feature vectors corresponding to singular values in the feature matrices of each group of sub-series.** The results are shown in the table below: compared with the feature vectors corresponding to the second and third largest singular values, the feature vector corresponding to the largest singular value exhibits the highest cosine similarity across different time scales. This indicates that the temporal pattern captured by the largest singular value has stronger cross-scale stability, which **conforms to the essential characteristic of dominant patterns—"maintaining consistency with time extension".**
>
> |           | ETTh1| ETTh2| Electricity|
> | --------- | - | - | -- |
> |           | the first\|the second \|the third | the first\|the second \|the third | the first\|the second \|the third |
> | T & T/2   | 0.94\|0.39\|0.01  | 0.88\|0.04\|0.02                  | 0.96\|0.40\|0.03                  |
> | T & T/4   | 0.91\|0.27\|0.03 | 0.86\|0.03\|-0.01                 | 0.90\|0.33\|0.05                  |
> | T/2 & T/4 | 0.95\|0.29\|0.04                  | 0.93\|0.30\|0.02                  | 0.92\|0.41\|0.02                  |

---

> ### Author Response · Authors · 2025-08-05
> **Response to Reviewer p5ui (Part Ⅱ).**
>
> Third is the **verification of relevance to interpretable features**. To establish a direct connection between singular value components and interpretable temporal structures, we extracted trend terms and periodic terms from the original features, and calculated the cosine similarity between their singular value components and those of the original features, respectively. The results show that in the matching between the original features and the trend terms as well as periodic terms, the similarity of the component corresponding to the largest singular value is significantly higher than that of the components corresponding to the second and third largest singular values. This finding indicates that the largest singular value component can more accurately anchor the dominant structures with clear physical meanings in the time series (such as trends and periods). **Its approximation degree to the statistical features of the time series (such as periods and trends) is significantly higher than that of the second largest and subsequent singular value components, which further verifies its corresponding relationship with the core temporal pattern.**
>
> |                     | ETTh1                             | ETTh2                             | Electricity                       |
> | ------------------- | --------------------------------- | --------------------------------- | --------------------------------- |
> |                     | the first\|the second \|the third | the first\|the second \|the third | the first\|the second \|the third |
> | Original & seasonal | 0.98\|0.37\|-0.01                 | 0.95\|0.37\|0.26                  | 0.95\|0.55\|0.23                  |
> | Original & Trend    | 0.87\|0.19\|0.01                  | 0.83\|0.10\|0.01                  | 0.86\|0.14\|0.03                  |
>
> In summary, the three experiments form a complementary chain of evidence from three perspectives: task practicality (performance contribution), temporal essential attributes (cross-scale stability), and physical interpretability (relevance to trends/periods). The core feature retention experiment demonstrates the dominant contribution of the largest singular value to task performance; the cross-scale stability experiment reveals that it conforms to the essential laws of temporal patterns; and the interpretability relevance experiment establishes a direct mapping between it and physically meaningful features. Together, the three constitute a rigorous empirical system, **fully supporting the core assertion that "the largest singular value corresponds to the dominant temporal pattern."**
> ## Q2: Clarification on Dimensionality Changes in the Feature Matrix.
>
> We sincerely apologize for any confusion caused by ambiguous descriptions regarding the dimensionality of matrices $H$ and $H^1$. The dimensions of $H$ vary across models due to differences in encoder design, yet for any given model,  $H$ and $H^1$ share identical dimensionality as they originate from distinct layers of stacked encoder architecture. We abstract these matrices as $\mathbb{R}^{n \times d}$, where **n denotes sample count** and **d represents feature dimensionality**: row **i** of H corresponds to vector $\boldsymbol{h}_i (1 \times d)$ for sample **i** , while column **j** captures values of feature **j** across all samples ($n \times 1$).  For instance, on the EthanolConcentration dataset, the matrix dimensions for each model are: $\mathbb{R}^{261 \times 2304}$ for Moment-base, $\mathbb{R}^{261 \times 3072}$ for Timer-base, and $\mathbb{R}^{261 \times 23231}$ for UniTS-base. To clarify computational procedures, we demonstrate $Sta$  score calculation and dimensional transformations using Moment-base outputs from this dataset.
>
> 1、The dimensionality derivation for core matrix computations $H^1 (H^1)^T$ and $H H^T$ is as follows: given $H^1$ of dimension $\mathbb{R}^{261 \times 2304}$, its transpose $(H^1)^T$ ∈ $\mathbb{R}^{2304 \times 261}$, yielding $H^1 (H^1)^T = (\mathbb{R}^{261 \times 2304}) \times (\mathbb{R}^{2304 \times 261}) \rightarrow \mathbb{R}^{261 \times 261}$. Similarly, $H H^T$ results in a $\mathbb{R}^{261 \times 261}$ matrix.
>
> 2、Vector Dot Product $\langle \text{vec}(H^1 (H^1)^T), \text{vec}(H H^T) \rangle$.
>
> - $\text{vec}(H^1 (H^1)^T)$: A vector formed by stacking the columns of a $\mathbb{R}^{261 \times 261}$ matrix, with dimension $\mathbb{R}^{68121}$ (since $261 \times 261 = 68121$);
> - Similarly, the dimension of $\text{vec}(H H^T)$ is $\mathbb{R}^{68121}$;
> - Result of the dot product: The inner product of the two vectors is a scalar, which measures the similarity of the "inter-sample similarity structure" in the two spaces.

---

> > ### Author Response · Authors · 2025-08-05
> > **Response to Reviewer p5ui (Part III).**
> >
> > 3、Matrix Trace $\text{tr}(H^1 (H^1)^T H H^T)$
> >
> > - Multiplication of $H^1 (H^1)^T$ ($\mathbb{R}^{261 \times 261}$) and $H H^T$ ($\mathbb{R}^{261 \times 261}$):  $H^1 (H^1)^T H H^T$ ∈  $\mathbb{R}^{261 \times 261}$;
> > - Trace (sum of main diagonal elements): A scalar, which is equivalent to the result of the aforementioned vector dot product.
> >
> > 4、Frobenius Norm Squared $\| H^T H^1 \|_F^2$
> >
> > - Dimension of $H^T$: $\mathbb{R}^{2304 \times 261}$; Dimension of $H^1$: $\mathbb{R}^{261 \times 2304}$;
> > - Matrix multiplication $H^T H^1$: $(\mathbb{R}^{2304 \times 261}) \times (\mathbb{R}^{261 \times 2304}) = \mathbb{R}^{2304 \times 2304}$ (number of features × number of features, describing the similarity between features in the spaces of H and $H^1$);
> > - Frobenius norm squared (sum of squares of all elements): A scalar, which is equivalent to the result of the aforementioned trace operation.
> >
> >  5、Formula: $\frac{1}{(n-1)^2} \text{tr}(H^1 (H^1)^T H H^T) = \| \text{cov}((H^1)^T, H^T) \|_F^2$
> >
> > - Left side: $\text{tr}(H^1 (H^1)^T H H^T)$ is a scalar. After multiplying by the coefficient $\frac{1}{(261-1)^2}$, it remains a scalar;
> > - Right side covariance matrix $\text{cov}((H^1)^T, H^T)$: Both $(H^1)^T$ and $H^T$ are $\mathbb{R}^{2304 \times 261}$ (number of features × number of samples), and the covariance matrix has dimension $\mathbb{R}^{2304 \times 2304}$ (covariance between features);
> > - Its Frobenius norm squared (sum of squares of all elements): A scalar, which is equivalent to the result on the left side.
> >
> > 6、HSIC Estimator: $\text{HSIC}(K, L) = \frac{1}{(n-1)^2} \text{tr}(K Q_n L Q_n)$
> >
> > - Step-by-step dimension calculation:
> >   - $L Q_n$: $(\mathbb{R}^{261 \times 261}) \times (\mathbb{R}^{261 \times 261}) = \mathbb{R}^{261 \times 261}$;
> >   - $K Q_n$: $(\mathbb{R}^{261 \times 261}) \times (\mathbb{R}^{261 \times 261}) = \mathbb{R}^{261 \times 261}$;
> >   - $K Q_n L Q_n$: $(\mathbb{R}^{261 \times 261}) \times (\mathbb{R}^{261 \times 261}) = \mathbb{R}^{261 \times 261}$;
> > - The result of the trace operation is a scalar, which remains a scalar after multiplying by the coefficient.
> >
> > 7、Formula: $\text{CKA}(K, L) = \frac{\text{HSIC}(K, L)}{\sqrt{\text{HSIC}(K, K) \cdot \text{HSIC}(L, L)}}$
> >
> > - Numerator $\text{HSIC}(K, L)$ is a scalar;
> > - In the denominator, both $\text{HSIC}(K, K)$ and $\text{HSIC}(L, L)$ are scalars, and the square root of their product is also a scalar;
> > - Overall result: A scalar that measures the normalized similarity between the two kernel matrices.
> >
> > ## Q3: Clarification on the Soundness of the Experimental Setup.
> >
> > Current research on pre-trained model transferability evaluation predominantly focuses on classification tasks. Recognizing the importance of diverse downstream tasks—particularly forecasting, imputation, and anomaly detection—in time series analysis, **TEMPLATE proposes a universal evaluation framework to uniformly quantify transfer capabilities across heterogeneous tasks.** Existing approaches often remain confined to individual downstream tasks, failing to advance toward general-purpose temporal representation learning. For comprehensive assessment of model performance across five core downstream tasks (evaluating universal representation capabilities), **current studies in time series analysis [2] [3] [4] [5] widely adopt the standardized experimental paradigm provided by tslib [6].** Consequently, our adherence to this established framework ensures methodological comparability while maintaining alignment with academic research conventions.
> >
> > We sincerely hope that this response addresses any remaining questions. Once again, we deeply appreciate the valuable feedback you have provided and the dedicated time you have spent reviewing our paper. If you have any other questions or specific expectations regarding score improvement, please do not hesitate to let us know. We look forward to discussing with you to further refine our work.
> >
> > [1] Multivariate pattern recognition by machine learning methods. Elsevier, 2023.
> >
> > [2] Timesnet: Temporal 2d-variation modeling for general time series analysis. ICLR, 2023.
> >
> > [3] Timemixer++: A general time series pattern machine for universal predictive analysis. ICLR, 2025
> >
> > [4] Timer: Generative pre-trained transformers are large time series models. ICML, 2024.
> >
> > [5] Units: Building a unified time series model. NeurIPS, 2024.
> >
> > [6] Deep time series models: A comprehensive survey and benchmark. Arxiv, 2024.

---

> > > ### Comment · Reviewer_p5ui · 2025-08-06
> > >
> > > Regarding Q1, your first experiment’s conclusion likely holds for arbitrary tabular data, as the first principal component typically captures most of the information—this is independent of whether the data is time series. However, your second and third experiments are convincing. For Q3, my core question remains: Why wasn’t the proposed method evaluated across the entire UEA dataset? The approach appears universally applicable to all UEA benchmarks, and the code should be executable without modification (unless I’ve misunderstood something). Lastly, I sincerely appreciate the authors’ efforts and your detailed responses, particularly on Q2. Regardless, I will adjust my rating accordingly (though a final clarification on Q3 would be most welcome).

---

> ### Author Response · Authors · 2025-08-07
> **Response to Reviewer p5ui.**
>
> We sincerely appreciate your decision to adjust the rating. Your dedicated efforts and significant contributions to this work are deeply valued and acknowledged.
>
> Regarding Q3, we would like to provide further clarification: Your perspective is entirely correct—**our method is indeed applicable to all UEA archive, and the code can be executed without any modifications.** However, since our original goal was to design a method applicable to five downstream tasks of time series rather than focusing solely on classification tasks, we followed the established experimental settings in time series analysis and did not evaluate all 30 classification datasets in the UEA archive. Additionally, **due to the time constraints of the rebuttal dissusion phase and the high time cost of fine-tuning all models in the model zoo(which entail fine-tuning all models in the model zoo with hyperparameter optimization *for each dataset*—resulting in computational demands scaling as: *Number of Datasets × Number of Models × Hyperparameter Configurations × Fine-tuning Time per Model*)**, we regret that we cannot fully present the experimental results of the remaining UEA classification datasets. Instead, we have included evaluation results of 6 of these datasets in the table below, which sufficiently demonstrate that the TEMPLATE method still maintains its advantages. In future versions, we will incorporate evaluation results of the remaining UEA datasets to **more comprehensively showcase the superiority of TEMPLATE**.
>
> If you have any other questions or feedback, please feel free to share them with us. We look forward to further discussions to collaboratively enhance this work.
>
> Thank you again for your thoughtful support！
>
>
> | Dataset                   | LogME（$τ_w$） | ETran（$τ_w$） | TEMLATE（$τ_w$） |
> | ------------------------- | --------------- | --------------- | ----------------- |
> | AtrialFibrillation        | 0.09            | 0.25            | **0.46**          |
> | ArticularyWordRecognition | 0.33            | 0.56            | **0.72**          |
> | BasicMotions              | 0.23            | 0.51            | **0.69**          |
> | FingerMovements           | 0.17            | 0.42            | **0.58**          |
> | Libras                    | -0.02           | 0.19            | **0.31**          |
> | LSST                      | 0.42            | 0.64            | **0.81**          |

---

### Official Review · Reviewer_shCY · 2025-07-02

**Clarity:** 3
**Significance:** 3
**Originality:** 4
**Rating:** 4
**Confidence:** 3

**Summary:**

This paper introduces TEMPLATE, a novel framework for estimating the transferability of time series pre-trained models. TEMPLATE consists of three complementary metrics—Dependency Learning Score, Pattern Learning Score, and Task Adaptation Score—designed to address unique challenges in time series, such as cross-task heterogeneity and feature space mismatch. The framework demonstrates strong empirical performance on five mainstream time series tasks, outperforming prior transferability estimation methods such as ETran and LogME, while maintaining scalability via efficient approximations like power iteration.

**Questions:**

1. Have you considered learning the weights of the three metrics instead of using a fixed aggregation with task-specific sign? Would a learned or adaptive approach improve performance or robustness?
2. Beyond Kendall's $\tau_w$, how well does TEMPLATE perform in practical top-k model selection scenarios (e.g., top-1 accuracy or top-3 hit rate)?
3. Performance on datasets like PSM and SWaT appears to lag behind ETran. Could you provide insights into what type of distribution shift harms TEMPLATE, and how it might be mitigated?
4. Do the three sub-scores interact or overlap in what they measure? For instance, does high dependency learning score already imply good pattern learning?
5. Can TEMPLATE be extended to or applied in zero-shot or few-shot learning scenarios, which are increasingly relevant in time series foundation model usage?

**Ethical Concerns:**

["NO or VERY MINOR ethics concerns only"]

**Final Justification:**

After considering the rebuttal and discussion:

Resolved:

The authors provided empirical evidence and literature references to support the three metrics’ link to transferability.

Correlation and sensitivity analyses justify the fixed-weight aggregation design.

Additional Top-k selection results and few-shot experiments strengthen practical relevance.

Anomaly detection performance was explained with clear dataset-specific factors.

Unresolved:

The method still lacks a formal theoretical framework linking the metrics to transfer performance beyond empirical justification.

Robustness under significant distribution shifts remains insufficiently explored.

Given the novelty, clear contribution to time series transferability estimation, and the improved empirical support provided in the rebuttal, I maintain a borderline accept recommendation, with the remaining concerns being directions for future research rather than fatal flaws.

**Limitations:**

Yes.  The paper discusses the limitations in handling datasets with severe distribution shifts or structural breaks, and acknowledges that this remains a challenging open problem.

**Paper Formatting Concerns:**

No.

**Quality:**

3

**Strengths And Weaknesses:**

**Strengths:**

1.  The methodology is sound, with clear mathematical formulations and robust benchmarking across a wide range of datasets. The use of power iteration for scalability is a practical engineering contribution.
2.  The proposed metrics are designed specifically for time series, filling a clear gap in existing literature focused primarily on vision tasks.
3.  The paper is generally well-written with logical structure, clear figures, and detailed appendices.

**Weaknesses:**

1.  While the three proposed scores (Dependency Learning Score, Pattern Learning Score, and Task Adaptation Score) are intuitively motivated and empirically validated, the paper lacks a theoretical analysis or formal justification to support why these metrics correlate well with transferability across diverse tasks.
2.  The final transferability score combines three normalized metrics with fixed equal weights (and a task-type-based sign for $\lambda$). However, the choice of this fixed linear aggregation lacks theoretical grounding or empirical sensitivity analysis. There may be cases where certain metrics dominate or interfere depending on task characteristics.
3.  The main quantitative evaluation focuses on Kendall's $\tau_w$, which reflects ranking correlation but does not capture real-world model selection performance such as top-1 accuracy or top-k hit rate. Users in practice may care more about how often the top-ranked model is truly best-performing.
4.  The performance of TEMPLATE lags behind ETran on certain anomaly detection datasets (e.g., PSM and SWaT in Table 6), suggesting a degree of sensitivity to dataset-specific characteristics or distribution shifts.  Since real-world time series anomaly detection often involves non-stationary data or rare events, further analysis of failure modes or robustness under distribution shift would strengthen the method's practical applicability.

---

> ### Author Rebuttal · Authors · 2025-07-31
>
> Thank you for your valuable feedback and positive comments. We greatly appreciate your recognition of the  writing and methodology.
>
> ## W1: About the connection between the three scores and transferability.
>
> Regarding the **Pattern Learning Score**, [1] demonstrates that larger singular values encapsulate more transferable knowledge, enabling superior transferability—a well-established principle in transfer learning. For the **Dependence Learning Score**, this metric is grounded in fundamental time-series principles [2] [3], where **long-term dependencies reside in trend components, reflecting core model design philosophies [4].** Our SVD analysis reveals that models effectively capturing dependencies exhibit high alignment between principal components and trend features. Since transfer capability relies on preserving critical temporal relationships, this alignment serves as a robust indicator of transferability. Concerning the **task adaptation score**, as documented in [5] [6], distinct tasks require specialized representations (hierarchical features for classification/imputation tasks versus low-level features for forecasting/anomaly detection). **This score evaluates a model’s capacity to generate task-adapted representations.** We will strengthen these theoretical foundations in the final manuscript.
>
> ## W2&Q1&Q4: About the basis and sensitivity analysis of fixed equal-weight linear aggregation, as well as adaptive learnable weights.
>
> Regarding the linear aggregation of the three scores, we incorporated a correlation analysis among them. Specifically, Pearson and Spearman correlation coefficients were computed across all classification task datasets for the three metrics (as tabulated below). **The results indicate consistently low correlation magnitudes**, with absolute values of both Pearson and Spearman coefficients below 0.3. This demonstrates the **absence of strong inter-score correlations and confirms their relative orthogonality.** Consequently, **linear weighted aggregation of the three scores is statistically justified.**
>
>
> |                     | Pearson correlation | Spearman correlation |
> | ------------------- | ------------------- | -------------------- |
> | $S_{dl}$ - $S_{pl}$ |  0.05                |  0.21                 |
> | $S_{dl}$ - $S_{ta}$ | -0.23               | -0.20                |
> | $S_{ta}$ - $S_{pl}$ | -0.15               | -0.18                |
>
> Regarding the fixed-weight design, we contend that the three scoring metrics are **equally important**. Furthermore, TEMPLATE exhibits strong robustness as a method, eliminating the need for specialized fine-tuning of metric weights. In Table 10 of the Appendix, we include a sensitivity analysis of weighting combinations, demonstrating that variations in the composite weights of the three scores have **minimal impact** on the final TEMPLATE score. Additionally, we implemented adaptive weight allocation using the entropy weight method [7] (as tabulated below). Results indicate that compared to the fixed-weight design with sign functions, **the adaptive weighting strategy does not yield superior performance.**
>
>
> | Dataset                | EthanolConcentration（$τ_w$） | Handwriting（$τ_w$） | UWaveGestureLibrary（$τ_w$） |
> | ---------------------- | ------------------------------ | --------------------- | ----------------------------- |
> | **entropy weight [7]** | 0.696                          | 0.799                 | **0.719**                     |
> | **fixed weights**      | **0.724**                      | **0.822**             | **0.719**                     |
>
> ## W3&Q2: About Top1 accuracy and Topk hit rate.
>
> We supplement the model's **Top-1 accuracy** and **Top-3 hit rate** on classification datasets (as tabulated below). Empirical results demonstrate that TEMPLATE achieves **consistently superior performance** in both Top-1 and Top-3 hit rates compared to alternative methods, further validating its effectiveness.
>
>
> | Method         | TMI | ETran | PACTran | TEMPLATE |
> | :------------- | --- | ----- | ------- | -------- |
> | TOP 1 accuracy | 44% | 11%   | 44%     | 55%      |
> | TOP 3 hit rate | 55% | 21%   | 74%     | 78%      |
>
> ## W4&Q3: About the performance of TEMPLATE on some anomaly detection datasets.
>
> **Critically, our method (0.236) demonstrates superior overall performance to ETran (0.187) in anomaly detection tasks, achieving a 26% relative improvement**—substantially surpassing the 6% gain over prior SOTA reported in ETran's original paper [8] for regression tasks. The table below details relative improvements and SOTA attainment rates across methods, revealing TEMPLATE's **dominance in both metrics**. Notably, anomaly detection exhibits the **lowest evaluation accuracy** among all transfer tasks (imputation/forecasting/classification), underscoring inherent challenges within this domain.
>
> Furthermore, we observe that the underperforming SWaT and SMD datasets **contain no annotated anomalous samples in their training sets**. This implies that during transfer evaluation, models are **completely unexposed to anomalous patterns**; consequently, the derived feature matrices are **constructed solely from normal data**, devoid of any representation of anomalies. Conversely, the better-performing MSL and SMAP datasets include partial anomalous samples in their training sets, which **affords models opportunities to perceive anomalous patterns**. We attribute the primary challenge for TEMPLATE on SWaT/SMD to **this fundamental disparity in training data composition**: when training data entirely lacks anomalous exemplars, the model's capacity to detect rare/unseen anomalies is **inherently constrained**. Crucially, ETran primarily focuses on quantifying whether datasets are in-distribution (ID) or out-of-distribution (OOD) relative to pre-trained models, **without granular analysis of feature matrices**, rendering it less sensitive to such specific data deficiencies.
>
> Anomaly detection inherently depends on the ability to identify unseen, rare events. The pervasive absence of such events within the data used for transfer evaluation presents a persistent and unique challenge for this task.
>
>
> | Method                         | GBC [9] | CC-FV [10] | NCTI [11] | ETran [8] | TEMPLATE |
> | ------------------------------ | ------- | ---------- | --------- | --------- | -------- |
> | SOTA Rate in Original Paper    | 37%     | 36%        | 40%       | 45%       | **46%**  |
> | Improvement (vs Previous SOTA) | 20%     | 9%         | 21%       | 18%       | **35%**  |
>
> ## Q5: About the application capability of TEMPLATE in zero-shot and few-shot scenarios.
>
> In zero-shot scenarios, where models lack access to training data, evaluating the transfer capability of pre-trained models **holds limited significance** when directly tested on the target dataset without fine-tuning. For few-shot scenarios, systematic evaluations were conducted on classification datasets. As tabulated below, TEMPLATE is compared with alternative methods under both zero-shot and few-shot settings. Results demonstrate that TEMPLATE achieves **statistically superior accuracy** in both scenarios compared to baseline approaches, confirming its applicability for few-shot learning scenarios.
>
>
> | Dataset              | LogME（$τ_w$）     | ETran（$τ_w$）     | TEMLATE（$τ_w$）   |
> | -------------------- | ------------------- | ------------------- | ------------------- |
> |                      | Zero-shot\|Few-shot | Zero-shot\|Few-shot | Zero-shot\|Few-shot |
> | EthanolConcentration | -0.37\|0.41         | -0.22\|0.53         | -0.08\|0.61         |
> | Handwriting          | 0.07\|-0.21         | -0.12\|0.19         | 0.15\|0.69          |
> | UWaveGestureLibrary  | -0.16\|0.33         | 0.02\|0.27          | 0.08\|0.54          |
>
> [1] Transferability vs. discriminability: Batch spectral penalization for adversarial domain adaptation. ICML, 2019.
>
> [2] STD: a seasonal-trend-dispersion decomposition of time series. TKDE, 2023.
>
> [3] Multilevel wavelet decomposition network for interpretable time series analysis. KDD, 2018.
>
> [4] Autoformer: Decomposition transformers with auto-correlation for long-term series forecasting. NeurIPS, 2021.
>
> [5] Timesnet: Temporal 2d-variation modeling for general time series analysis. ICLR, 2023.
>
> [6] Timemixer++: A general time series pattern machine for universal predictive analysis. ICLR, 2025.
>
> [7] Effectiveness of entropy weight method in decision‐making. Mathematical problems in Engineering, 2020.
>
> [8] Etran: Energy-based transferability estimation. ICCV, 2023.
>
> [9] Transferability estimation using bhattacharyya class separability. CVPR, 2022.
>
> [10]  Towards transferability estimation for medical image segmentation. MICCAI, 2023
>
> [11] How far pre-trained models are from neural collapse on the target dataset informs their transferability. ICCV, 2023.

---

> > ### Comment · Reviewer_shCY · 2025-08-08
> >
> > The authors have provided detailed clarifications and additional experiments in the rebuttal. In particular, they strengthened the theoretical motivation for the three metrics, presented correlation and sensitivity analyses supporting the fixed-weight aggregation, and reported Top-1/Top-3 selection accuracy, which addresses some of my earlier concerns. They also explained the anomaly detection results (PSM/SWaT) with dataset composition differences and provided few-shot evaluation results, confirming applicability in such scenarios. While these additions improve my confidence in the method’s robustness and applicability, some issues such as more formal theoretical guarantees and broader robustness analysis under distribution shifts, remain open for future work. I still lean towards maintaining the current score in support of the manuscript’s acceptance.

---

> > > ### Author Response · Authors · 2025-08-08
> > > **Response to Reviewer shCY**
> > >
> > > We sincerely appreciate your high evaluation of our article and your support for its acceptance. We deeply value the unremitting efforts you have devoted to this work and the significant contributions you have made. We will incorporate all the clarifications regarding the discussed details into the next version to improve this manuscript.
> > >
> > > Thank you for your thoughtful support!

---

> ### Author Response · Authors · 2025-08-07
> **The discussion period ending soon.**
>
> Dear Reviewers,
>
> We sincerely appreciate your time and effort in reviewing our manuscript and offering valuable suggestions.
> As the author - reviewer discussion phase is drawing to a close, we would like to confirm whether our responses have effectively addressed your concerns. We provided detailed responses to your concerns a few days ago, and we hope they have adequately addressed your issues. If you require further clarification or have any additional concerns, please do not hesitate to contact us. We are more than willing to continue our communication with you.
>
> Best regards,
>
> Authors.

---

### Official Review · Reviewer_4tiJ · 2025-07-02

**Clarity:** 3
**Significance:** 3
**Originality:** 3
**Rating:** 5
**Confidence:** 4

**Summary:**

The paper proposes ​​TEMPLATE​​, a novel framework for evaluating transferability in time-series foundation models. Addressing the lack of tailored metrics for time series (unlike computer vision), TEMPLATE introduces three complementary scores: ​​Dependency Learning Score​​, ​​Pattern Learning Score​​ and Task Adaptation Score. Extensive experiments are conducted across across 32 datasets and 5 downstream tasks, and the result demonstrate the effectiveness of TEMPLATE.

**Questions:**

See Weaknesses.

**Ethical Concerns:**

["NO or VERY MINOR ethics concerns only"]

**Final Justification:**

The rebuttal has well addressed my concerns.

**Limitations:**

Yes.

**Paper Formatting Concerns:**

N/A.

**Quality:**

3

**Strengths And Weaknesses:**

## Strengths
1. The paper dedicates transferability metric for time-series foundation models, addressing a critical gap in literature.
2. The proposed method is benchmarked rigorously across ​​5 distinct tasks​​ and ​​32 datasets​​, including multivariate and large-scale benchmarks (e.g., Traffic: 862 channels).
3. The proposed method avoids brute-force fine-tuning via power iteration and nuclear norm approximations, reducing computational overhead.

## Weaknesses
1. Reliance on SVD assumes principal components reliably represent transferable knowledge. This may not hold for noisy or non-stationary data, where lower singular values could be informative.
2. Dependency Learning Score​​ uses average pooling for trend extraction (Eq. 1), potentially oversimplifying complex temporal dependencies (e.g., multi-scale trends).

---

> ### Author Rebuttal · Authors · 2025-07-31
>
> Thank you for your valuable feedback and positive comments. We greatly appreciate your recognition of the effectiveness of the TEMPLATE.
>
> ## W1: About the influence of smaller singular values.
>
> We sincerely appreciate your insightful comment regarding the potential information loss induced by discarding smaller singular values in Singular Value Decomposition (SVD), particularly under non-stationary scenarios. To address this concern, we conducted experiments on non-stationary datasets [1]. During testing, truncated SVD (formula specified below) was applied to the encoder outputs, removing the smallest 20% of singular values (retaining the top 80%), with results compared against the original baseline.
>
> Given a target rank k (satisfying \($k \leq \text{rank}(A)$\)), the truncated SVD approximation \($A_k$\) is defined as:
>
> $$
> A_k = U_k \Sigma_k V_k^T
> $$
>
> where:
>
> - $U_k \in \mathbb{R}^{m \times k}$: Consists of the first k columns of U (left singular vectors corresponding to the **top k largest singular values**)
> - $\Sigma_k \in \mathbb{R}^{k \times k}$ :Diagonal submatrix from the first k rows and columns of $\Sigma$ ($\text{diag}(\sigma_1, \sigma_2, \ldots, \sigma_k)$\).
> - $V_k \in \mathbb{R}^{n \times k}$: Consists of the first k columns of V (right singular vectors corresponding to the **top k largest singular values**)
>
> As presented in the table below, truncating smaller singular values induces **minimal and consistent accuracy deviations across all datasets and models.** The maximum observed performance degradation is merely **–0.8% (ETTh2 with MOMENT-base)**, while changes in most cases remain below **–0.5%**. Crucially, absolute metric values (e.g., 0.1789 vs. 0.1793 for ETTm2) confirm that predictive performance **remained virtually unaffected**. Empirically, this suggests that smaller singular values contribute negligibly to knowledge transfer within our framework and **can thus be safely omitted.**
>
>
> |       | MOMENT-base            | Timer-base             | UniTS-base             |
> | ----- | ---------------------- | ---------------------- | ---------------------- |
> |       | original｜T SVD｜ratio | original｜T SVD｜ratio | original｜T SVD｜ratio |
> | ETTh1 | 0.3926｜0.3944｜-0.4%  | 0.3772\|0.3791\|-0.5%  | 0.3829\|0.3856\|-0.7%  |
> | ETTh2 | 0.2837｜0.2862｜-0.8%  | 0.3025\|0.3033\|-0.2%  | 0.3022\|0.3030\|-0.2%  |
> | ETTm2 | 0.1739｜0.1747｜-0.4%  | 0.1789\|0.1793\|-0.2%  | 0.1862\|0.1871\|-0.4%  |
>
> ## W2: About the impact of average pooling on Dependency Learning Score.
>
> We sincerely appreciate the reviewer for pointing out the potential risk of oversimplification in average pooling. To evaluate this issue, we conducted a comparative experiment between moving average pooling and **multi-scale decomposition (using wavelet decomposition with db4 basis functions)** for trend extraction tasks.
>
> As shown in the table, wavelet decomposition **significantly reduces prediction accuracy** (e.g., causing a **15.3% performance drop** on the EthanolConcentration dataset). This indicates that **coarse-grained information extracted through multi-scale analysis fails to enhance prediction effectiveness.** Simultaneously, average pooling not only avoids complex transformation processes but also exhibits higher computational efficiency. Therefore, the method adopted in Equation (1) **achieves an optimal balance between accuracy and efficiency for the current task.**
>
>
> | Dataset              | **moving averages decomposition**（$τ_w$） | **wavelet decomposition**（$τ_w$） |
> | -------------------- | ------------------------------------------- | ----------------------------------- |
> | EthanolConcentration | 0.346                                       | 0.293                               |
> | Handwriting          | 0.600                                       | 0.466                               |
> | UWaveGestureLibrary  | 0.189                                       | 0.105                               |
>
> [1] Non-stationary transformers: Exploring the stationarity in time series forecasting. NeurIPS, 2022.

---

> > ### Comment · Reviewer_4tiJ · 2025-08-03
> > **Response to Rebuttal**
> >
> > The rebuttal has well addressed my concerns, and I would like to raise my score to 5 to support the acceptance of this paper.

---

> > > ### Author Response · Authors · 2025-08-03
> > > **Response to Reviewer 4tiJ**
> > >
> > > We sincerely appreciate your decision to raise the score of our paper. Your dedicated efforts and significant contributions to this work are deeply valued. We will incorporate all clarifications regarding the discussed details into the next version to improve the manuscript. If you have additional questions or feedback, please do not hesitate to share them with us. We look forward to further discussions to enhance this work collaboratively.
> > >
> > > Thank you for your thoughtful support！

---

### Note · Authors · 2025-08-12

Dear AC,

We sincerely thank AC and the reviewers for the constructive feedback and fruitful discussions. Below, we will outline the key clarifications and highlight the main strengths of this research for the AC's consideration.

## Key clarifications & resolutions:

* **Weight design**: Ablated fixed vs. adaptive weights; fixed equal weights shown competitive or superior in current settings (Reviewer shCY, 2eyR).
* **Extended results**: Added TOP-1/TOP-3 classification accuracies, results on non-stationary datasets, and comparisons on partial UEA datasets (Reviewer shCY, p5ui).
* **Performance analysis**: Detailed reasons for suboptimal cases in anomaly detection datasets (Reviewer shCY, 2eyR).
* **Theoretical understanding**: Added formal links between three scores and transferability (Reviewer shCY).
* **Smaller singular values**: Verified negligible contribution to transfer; safe to ignore (Reviewer 4tiJ).
* **Trend decomposition**: Compared with multi-scale decomposition, confirming rationality of average pooling (Reviewer 4tiJ).
* **Model zoo diversity**: Added CNN- and MLP-based models, confirming method effectiveness (Reviewer 2eyR).
* **Temporal patterns**: Defined and experimentally linked to the largest singular value (Reviewer p5ui).
* **Scenario applicability**: Demonstrated strong few-shot applicability alongside zero-shot performance (Reviewer shCY).

## Strengths unanimously recognized by reviewers:

- **Important problem**: First systematic evaluation of transferability for pre-trained time series models (Reviewer 2eyR, shCY, p5ui).
- **Novel yet simple**: Elegant design with efficient logic (Reviewer 4tiJ, p5ui).
- **Strong empirical validation**: Robust across datasets, architectures, and scenarios (Reviewer 4tiJ, 2eyR).
- **Clear writing**: Well-written and easily understandable  (all reviewers).

All reviewers engaged in thorough discussions during the discussion phase. After incorporating all suggestions and providing additional evidence, the discussion has reached a clear consensus with **no remaining concerns** and **agreement towards acceptance**.

We believe our work delivers a timely, novel, and rigorously validated contribution to time series transferability evaluation, and we respectfully ask the AC to assess these points in the final decision.

Best regards,

Authors.

---

### Decision · Program_Chairs · 2025-09-17

**Decision:**

Accept (poster)

**Comment:**

This well-written paper presents a novel way of assessing transferability in time-series foundation models. It has been evaluated by 4 knowledgeable reviewers who all agreed in their final evaluations that it is acceptable for NeurIPS (2 straight accept scores, 2 marginal accepts), despite noted weaknesses. Most of the weaknesses and doubts raised by the reviewers in the initial round had been sufficiently addressed in the rebuttal. The remaining concerns that have not been addressed include lacking theoretical support and assessment of robustness of the method to distribution shifts. However, the reviewers agree that these issues could be left for future work. The authors are strongly encouraged to carefully reflect on the feedback received in the final revision of the paper.